# The Attack Means Nothing: Test-time Adversarial Defense Improves Zero-shot Adversarial Robustness for Medical Vision-Language Models

## Abstract

Vision-language models (VLMs), exemplified by CLIP, have achieved remarkable zero-shot generalization but remain highly vulnerable to imperceptible adversarial perturbations, posing significant safety threats, particularly in medical scenarios. In this paper, we first prove that VLMs are much more robust than adversarial attacks when faced with weak transformations. Building upon this insight, we propose the **T**he **A**ttack **Me**ans Nothing (TAME), a simple yet effective test-time defense paradigm for improving the zero-shot adversarial robustness of medical VLMs. We conduct comprehensive experiments on 11 medical datasets across 9 imaging modalities against three representative white-box attacks (PGD, C&W, and AutoAttack). The BiomedCLIP with a backbone of ViT-B/16 is utilized as the victim model. Extensive experiment results demonstrate that our TAME consistently outperforms other defense methods across all attack types, boosting the vanilla BiomedCLIP by $+47.47\%$ under PGD, $+46.73\%$ under C&W, and $+47.79\%$ under AutoAttack, while maintaining competitive clean accuracy. These significant improvements also suggest a potential risk of label leakage during attacks. Furthermore, our TAME is plug-and-play and can be integrated with other adversarially fine-tuned VLMs to enhance their defense capabilities. These findings support a practical and generalizable approach to deploying medical VLMs in clinical scenarios with the presence of adversaries. Codes will be available on GitHub.

## 1 Introduction

Recent advancements in vision-language models (VLMs) Zhao et al. (2025); Lai et al. (2024) have demonstrated significant success and potential for medical image analysis Koleilat et al. (2025); Stevens et al. (2024). Unlike traditional supervised learning focused on closed-set tasks, VLMs, such as Contrastive Language-Image Pre-training (CLIP) Radford et al. (2021a), enable the exploration of open-set visual concepts, yielding strong zero-shot generalization capabilities. Unfortunately, some studies Zhang et al. (2022); Zhao et al. (2023); Yin et al. (2023) reveal that adding even imperceptible adversarial perturbations to input images can severely degrade VLM's inference ability. This poses critical safety risks, especially in medical scenarios Dong et al. (2024), which may lead to serious misdiagnosis and hinder models from being deployed in real-world applications (see Figure 1).

Extensive research has explored adversarial training Chen et al. (2020) as an effective defense strategy, which can be broadly categorized into two categories: adversarial fine-tuning (AFT) Mao et al. (2023); Wang et al. (2024a); Schlarmann et al. (2024); Wang et al. (2024c) and adversarial prompt tuning (APT) Li et al. (2024); Zhang et al. (2024); Zhou et al. (2024); Wang et al. (2024b); Zhou et al. (2024). AFT methods aim to establish a min-max game between the VLM and an adversary, fine-tuning the pre-trained VLM on generated adversarial examples to achieve transferable robustness across downstream tasks. However, most of these methods require substantial computational resources and inevitably degrade the model's generalization to testing data from unseen distributions. APT methods attempt to train learnable textual or visual prompts by aligning adversarial image embeddings with corresponding text prompts while keeping the model backbone frozen. Al-

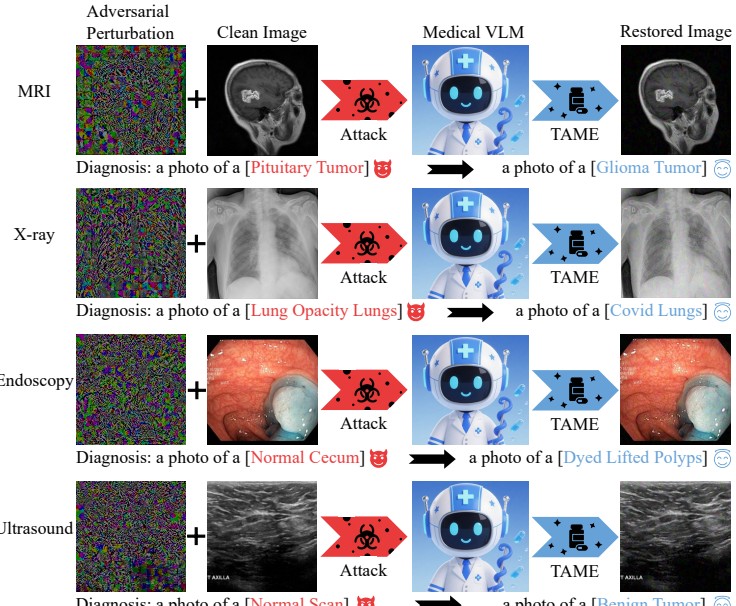

Figure 1: Adversarial attacks disturb model inference by adding imperceptible perturbations to the input image, leading to serious misdiagnosis. Our TAME enables the medical VLM to remain robust against adversarial attacks during inference without extra training on predefined adversarial data.

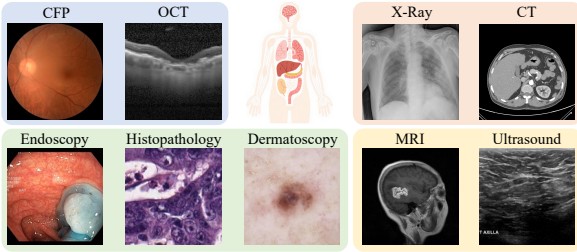

Figure 2: The medical imaging modalities used in this study. CFP: Color fundus photography. OCT: Optical coherence tomography. CT: Computed tomography. MRI: Magnetic resonance imaging.

though they reduce training costs, their effectiveness is constrained by predefined data distributions, limiting the adaptability to out-of-distribution environments. Consequently, achieving low-cost and effective adversarial robustness remains an open challenge.

Test-time adversarial defense (TAD) Alfarra et al. (2022); Pérez et al. (2021); Wu et al. (2021); Guo et al. (2018); Xing et al. (2025); Wang et al. (2025); Mao et al. (2021) has emerged as a promising paradigm to boost zero-shot adversarial robustness in a low-cost manner, as TAD requires only test data during the inference phase. Training-free TAD methods Pérez et al. (2021); Guo et al. (2018) assemble several image transformations to make it difficult for adversaries to circumvent the defense strategy. Training-based approaches Alfarra et al. (2022); Xing et al. (2025); Wu et al. (2021); Sheng et al. (2025); Mao et al. (2021) mainly focus on modifying the input image or training a prompt to counteract attacks. Despite their efforts, almost all existing TAD methods are designed for conventional networks like convolutional neural networks, with insufficient exploration of VLM. Furthermore, medical VLMs are typically utilized to process a wide range of modalities, as illustrated in Figure 2, posing a challenge to the defense method's generalizability across various modalities.

The key to addressing these issues is to identify commonalities of adversarial images to establish a general test-time defense paradigm for VLMs. In this paper, we first conduct a toy experiment on multiple datasets by applying several transformations to both clean and adversarial images. We observe that although transformations with large magnitude significantly disturb model predictions on

both images, this abnormal effect still appears on the adversarial images even under transformations of minor magnitude. We term this phenomenon **'semantic fragility'** of adversarial perturbations, which can be interpreted as these perturbations being highly specific to the corresponding input images. The VLM, trained on extensive and diverse data, exhibits inherent robustness to such minor transformations on clean images. In contrast, adversarial perturbations are over-fitted to both the specific input and the current model parameters, rendering the semantic content within the perturbed image embeddings highly susceptible to even slight alteration. Based on this observation, we propose **T**he **A**ttack **Me**ans Nothing (TAME), a test-time defense paradigm for medical VLMs. TAME counteracts adversarial attacks by training an adversarial restoration map for each adversarial image in a single iteration. Specifically, we first introduce the adversarial restoration map to the input image to produce the restored image and then minimize the KL divergence between the predicted probability distributions of each restored image and its transformed version. Through this training process, the trained adversarial restoration map learns to restore the model's invariance to minor transformations, thereby reinstating the inference capability of VLMs. Furthermore, the adversarial restoration map should also minimize its effect on clean images, thereby avoiding significant performance degradation induced by excessive image modification. To address this issue, we design a dynamic weighting mechanism that adaptively allocates weights according to the degree of semantic fragility exhibited by the input image. Comprehensive experiments are conducted across 11 medical classification datasets, including 9 modalities (see Figure 2), to evaluate TAME and other state-of-the-art methods against three typical adversaries (*i.e.*, Projected Gradient Descent PGD Madry et al. (2018), C&W Carlini & Wagner (2017), and AutoAttack Croce & Hein (2020)) that aim to maximize the classification loss in a white-box setting. Extensive experiment results demonstrate the effectiveness and superiority of our TAME across diverse scenarios.

The three key distributions of this paper are summarized as follows.

- We propose a simple yet effective method to enhance the zero-shot adversarial robustness of medical Vision-Language Models (VLMs), which can be utilized as a plug-and-play module without additional training.

- Based on observed commonalities in adversarial images, we propose TAME to protect VLMs against multiple attack types alongside a dynamic weighting mechanism maintaining performance on clean images.

- Extensive experiments on 11 medical classification datasets across 9 modalities demonstrate the superiority of our TAME over other existing defense methods.

## 2 RELATED WORK

### 2.1 ADVERSARIAL TRAINING

Adversarial training enhances the adversarial robustness of the model by training on the predefined adversarial samples, which can be broadly classified into adversarial fine-tuning Mao et al. (2023); Wang et al. (2024a); Schlarmann et al. (2024); Wang et al. (2024c) and adversarial prompt tuning Li et al. (2024); Zhang et al. (2024); Zhou et al. (2024); Wang et al. (2024b).

**Adversarial Fine-Tuning (AFT)** AFT improves the adversarial robustness by fine-tuning the VLM on adversarial samples generated by an adversary. Mao *et al.* Mao et al. (2023) fine-tuned the vision encoder of CLIP using adversarial contrastive learning with text-guided supervision on a small set of adversarial samples. Wang *et al.* Wang et al. (2024a) proposed a pre-trained model guided adversarial fine-tuning method, which distills the general knowledge from the original pre-trained model to the target model to mitigate the over-fitting. Schlarmann *et al.* Schlarmann et al. (2024) attempted to minimize the distance between the original and fine-tuned image embeddings during adversarial training to preserve the performance of the fine-tuned model on clean data.

**Adversarial Prompt Tuning (APT)** APT learns trainable prompts to maintain alignment under attack by exposing the model to adversarial samples while freezing the model parameters. Zhou *et al.* Zhou et al. (2024) presented to learn adversarially correlated text supervision by enhancing the consistency of multi-modal features and encouraging distinguishability between features of clean and adversarial data. Zhang *et al.* Zhang et al. (2024) aligned learnable text prompts with adversarial image embeddings to improve resistance against white-box and black-box adversarial attacks. Li *et*

*al.* Li et al. (2024) demonstrated the high sensitivity of both adversarial attacks and defenses to the specific text prompts used in VLMs and proposed to improve adversarial robustness by learning robust text prompts. Unlike these adversarial training methods, our TAME aims to achieve zero-shot adversarial robustness using only test data in a low-cost and general manner, improving the performance across various medical scenarios.

## 2.2 TEST-TIME ADVERSARIAL DEFENSE (TAD)

TAD aims to protect the pre-trained model from adversarial attacks in a low-cost manner during inference, including two branches: training-free and training-based methods. Training-free methods Pérez et al. (2021); Guo et al. (2018) typically refer to designing the image transformation strategy. Pérez *et al.* Pérez et al. (2021) proposed a transformation ensemble method achieving consistent improvements in adversarial robustness across datasets and adversaries while preserving clean data performance. Guo *et al.* Guo et al. (2018) found that total variance minimization and image quilting are effective against several attacks, particularly on the model trained on such transformation strategies. Training-based methods Alfarra et al. (2022); Xing et al. (2025); Wu et al. (2021); Wang et al. (2025); Mao et al. (2021) primarily prevent the model from adversarial attacks by modifying the input image or training a prompt. Alfarra *et al.* Alfarra et al. (2022) presented the anti-adversary layer to generate a perturbed input image in the opposite direction of the adversarial one to counter the attacks. Xing *et al.* Xing et al. (2025) maximized the classification loss on the test image to counterattack adversaries and prevented further counterattacking on clean data using a threshold. Sheng *et al.* Sheng et al. (2025) reformulated the marginal entropy objective to train the textual prompts and proposed a reliability-weighted ensembling strategy that aggregates information from trustworthy augmented views to enhance defense. Although TTC and R-TPT are designed for VLMs, they still exhibit limitations: 1) TTC requires a hyperparameter to distinguish clean and adversarial images during inference, leading to suboptimal performance in adversarial robustness when misclassifications occur. 2) R-TPT relies on low-entropy predictions for pointwise entropy minimization, however, adversarial images typically yield high-confidence but incorrect predictions, potentially reinforcing wrong decisions. In contrast, our TAME explores the inherent defect of adversarial attacks in VLMs and requires only one gradient backpropagation step. It employs a general training objective for both clean and adversarial images to achieve zero-shot adversarial robustness.

## 3 METHODOLOGY

### 3.1 PRELIMINARIES

**Zero-shot inference of CLIP.** Let $f_{\theta_v}$ and $f_{\theta_t}$ denote the CLIP's vision encoder and text encoder, respectively, where $\theta_v$ and $\theta_t$ are their corresponding parameters. Given an input image $I$ and a set of $k$ possible classes $C = \{c_1, c_2, ..., c_k\}$, $I$ can be classified in a zero-shot manner by computing the cosine similarity between the produced image embedding and the text embeddings of $C$ wrapped in a template(*e.g.*, "a photo of [CLASS]"). Specifically, the cosine similarity score between the image embedding and the text embedding of $i$-th class $c_i$ can be formulated as:

$$Cos_i(I) = \frac{f_{\theta_v}(I) f_{\theta_t}(c_i)^\top}{\|f_{\theta_v}(I)\| \cdot \|f_{\theta_t}(c_i)\|} \tag{1}$$

The probability that $I$ belongs to class $c_i$ is then calculated by the Softmax function:

$$p_i(I) = Softmax(Cos_i(I)) = \frac{e^{Cos_i(I)/\tau}}{\sum_{n=1}^{k} e^{Cos_n(I)/\tau}}, \tag{2}$$

where $\tau$ refers to the temperature. The predicted class corresponds to the highest probability: $\arg\max_i p_i(I)$. For clarity, we will utilize $P(I)$ to indicate the probability predictions of all classes.

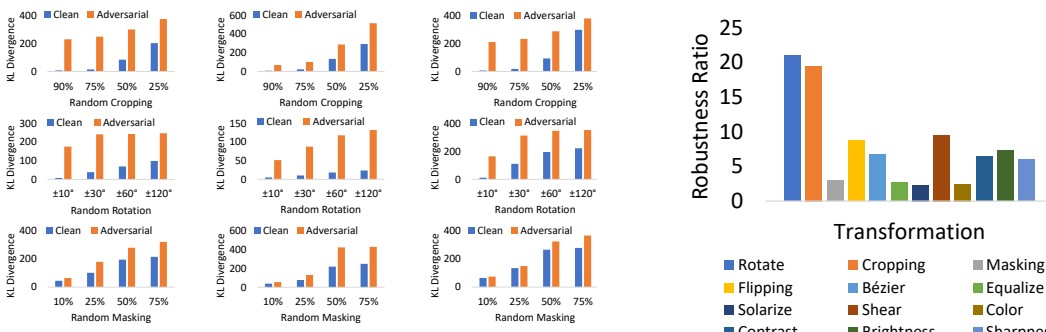

Figure 3: (a) Pipeline of training an adversarial perturbation map $\delta_p$ for a specific image $I$ with its corresponding label. (b) Pipeline of inference with an adversarial input image $I + \delta_p$. The black and red arrows indicate the data flow and gradient flow, respectively.

Figure 4: The KL divergence between BiomedCLIP's predictions before and after applying transformations across three datasets with various modalities.

Figure 5: Averaged robustness ratio of 12 transformation strategies calculated on 11 datasets.

**Adversarial attack.** We focus on three typical adversarial attack methods under a white-box setting: PGD Madry et al. (2018), C&W Carlini & Wagner (2017), and AutoAttack Croce & Hein (2020). This setting assumes that adversaries have complete access to the architecture and parameters of the victim model, enabling direct gradient-based attacks. The pipeline is shown in Figure 3. The adversary learns an adversarial perturbation to increase the divergence between text and image embeddings by maximizing an adversarial perturbation loss $\mathcal{L}_p$ (e.g., cross-entropy) as follows:

$$\delta_p = \underset{\|\delta\|_{\infty} \leq \epsilon_p}{\arg\max} \mathcal{L}_p(P(I + \delta), c_y) \tag{3}$$

where $\epsilon_p$ and $c_y$ denote the perturbation budget and the class label, respectively. The adversary then employs the victim VLM on the adversarial image $\hat{I} = I + \delta_p$ to obtain an incorrect prediction.

### 3.2 SEMANTIC FRAGILITY OF ADVERSARIAL PERTURBATIONS

In this study, we found that VLM is more robust than adversarial perturbations, and the perturbed image embeddings are semantically fragile and highly sensitive to minor alterations. To demonstrate this, we first performed toy experiments on three representative datasets from three modalities (MRI, dermoscopy, and X-ray) using three types of transformations: random cropping, random rotation, and random masking. Specifically, we measured the robustness by calculating the symmetrical KL divergence between predictions before and after applying transformations as follows:

$$r(I, I_t) = \mathcal{L}_{kl}(P(I), P(I_t)) + \mathcal{L}_{kl}(P(I_t), P(I)), \tag{4}$$

where $\mathcal{L}_{kl}$ indicates the KL divergence loss, and $I_t$ denotes the transformed image. PGD and BiomedCLIP Zhang et al. (2023) are utilized as the adversarial attack method and the victim model, respectively. As illustrated in Figure 4, weak transformations (e.g., $\pm10°$ random rotation and 90% random cropping) produce low KL divergence for clean images but high divergence for adversarial

ones, with this gap decreasing under stronger transformations. This phenomenon can be attributed to that adversarial perturbations are highly over-fitted to the specific image. However, this effect is dependent on the transformation type. In contrast, random masking has a minimal effect at low intensities, as it only alters a small portion of the adversarial perturbations.

To discuss the efficacy of various transformation strategies, we further defined a ratio $R = r(\hat{I}, \hat{I}_t)/r(I, I_t)$ to quantify the robustness discrepancy induced by a transformation $t$ between clean and adversarial images. A larger value of $R$ indicates a stronger discriminative capacity of $t$ in distinguishing adversarial images from clean ones. We calculated $R$ on 13 common transformation strategies and displayed the results in Figure 5. It shows that an effective transformation strategy yielding a high $R$ should satisfy two criteria: (1) it should modify the values and/or spatial positions of a majority of pixels, thereby amplifying its impact on adversarial images; and (2) it should apply minimal distortion to preserve high robustness on clean images. Therefore, we selected random cropping and random rotation, which exhibit high $R$ values, to construct our defense strategy. Comprehensive experiment results are provided in the Appendix.

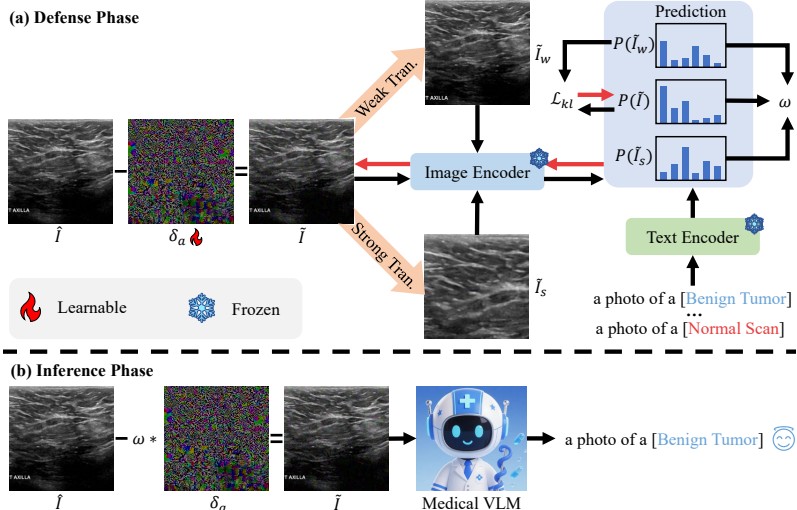

Figure 6: (a) Pipeline of our TAME training the adversarial restoration map $\delta_a$ for an adversarial image $\hat{I}$. (b) Pipeline of inference with the weighted restored image $\tilde{I}$. The black and red arrows indicate the data flow and gradient flow, respectively. 'Tran.': Abbreviation of 'Transformation'.

### 3.3 THE ATTACK MEANS NOTHING (TAME)

#### 3.3.1 ADVERSARIAL RESTORATION

In this paper, we specifically target the preservation of VLM's zero-shot inference robustness, where the defender has neither access to task-specific training data nor annotations for test samples. Based on the above observations (see Section 3.2), we proposed TAME, a simple yet effective method, as illustrated in Figure 6. For each input adversarial image $\hat{I}$, we introduced a learnable adversarial restoration map $\delta_a$, yielding the restored image $\tilde{I} = \hat{I} - \delta_a$, which is intended to approximate the original clean image $I$. Due to the inability to directly obtain $I$, we adopted a compromise approach that restores the model's strong robustness on weak transformations by posing the consistency constraint to train $\delta_a$ as follows:

$$\min_{\|\delta_a\|_\infty \leq \epsilon_a} \mathcal{L}_{kl}(P(\tilde{I}), P(\tilde{I}_w)), \tag{5}$$

where $\epsilon_a$ and $\tilde{I}_w$ indicate the defense budget and the weakly transformed $\tilde{I}$, respectively. Note that the $\mathcal{L}_{kl}(P(\tilde{I}_w), P(\tilde{I}))$ term is removed since $P(\tilde{I})$ approximates $P(\hat{I})$, which may provide limited supervision. This process to update $\delta_a$ can be approximated by PGD Madry et al. (2018):

$$\delta_a^1 = \prod(\delta_a^0 - \alpha \, sgn(\nabla_{\delta_a} \mathcal{L}_{kl}(P(\tilde{I}), P(\tilde{I}_w)))), \tag{6}$$

where $\alpha$ denotes the step-size, and the update step is fixed at $1$ in this study. The initial perturbation map $\delta_a^0$ is randomly sampled from a uniform distribution $U(-\epsilon_a, \epsilon_a)$.

### 3.3.2 WEIGHTING MECHANISM

In this section, we attempt to address the risk that directly applying $\delta_a$ to produce the restored image may degrade the performance on clean images, thereby hindering model deployments. An intuitive strategy is to leverage the divergent responses of clean and adversarial images to minor transformations to distinguish them. However, it is hard to determine a universal threshold due to the discrepancies among datasets. Recall the observation shown in Figure 4 that strong transformations with large magnitude induce high KL divergence values on both clean and adversarial images. Derived from this, such a high KL divergence value can be treated as an anchor to take the effect of normalization across datasets. We then devised a dynamic weight coefficient $\omega$ formulated by:

$$\omega = \frac{\mathcal{L}_{kl}(P(\tilde{I}), P(\tilde{I}_w))}{\mathcal{L}_{kl}(P(\tilde{I}), P(\tilde{I}_s))} \tag{7}$$

where $\tilde{I}_s$ indicates the strongly transformed $\tilde{I}$. Due to the discrepancy between clean and adversarial images, this ensures allocating larger weights to adversarial ones while avoiding excessive modification of clean ones. To further amplify the effect of $\omega$ on adversarial images and mitigate potential instability from excessively large weights, we also truncated $\omega$ using an empirically determined threshold value of 0.5, where values exceeding this threshold were clipped to 1. Then the model can perform inference on the weighted restored image as illustrated in Figure 6 (b). We summarize the algorithm of our TAME in Algorithm 1.

---

**Algorithm 1:** TAME Algorithm.

---

**Input:** Current test image $I$, pre-trained VLM (including $f_{\theta_v}$ and $f_{\theta_t}$), defense budget $\epsilon_a$, and step-size $\alpha$.
  1: $\delta_a^0 \sim U(-\epsilon_a, \epsilon_a)$.
  2: $\delta_a^1 = \prod(\delta_a^0 - \alpha sgn(\nabla_{\delta_a}\mathcal{L}_{kl}(P(I), P(I_w))))$.
  3: $\delta_a = \text{clamp}(\delta_a^1, -\epsilon_a, \epsilon_a)$.
  4: $\omega = \frac{\mathcal{L}_{kl}(P(I), P(I_w))}{\mathcal{L}_{kl}(P(I), P(I_s))}$.
  5: $\omega = \omega \cdot \mathbb{1}_{\omega \leq 0.5} + 1 \cdot \mathbb{1}_{\omega > 0.5}$
**Output:** $P(I - \omega \cdot \delta_a)$

---

## 4 EXPERIMENTS AND RESULTS

### 4.1 DATASETS AND METRIC

We conducted all experiments on the test sets of 11 diverse medical datasets spanning 9 imaging modalities: Computerized Tomography (CTKidney Islam et al. (2022)), Dermatoscopy (DermaM-NIST Codella et al. (2019); Tschandl et al. (2018)), Endoscopy (Kvasir Pogorelov et al. (2017)), Color Fundus Photography (RETINA Köhler et al. (2013); Porwal et al. (2018)), Histopathology (LC25000 Borkowski et al. (2019) and CHMNIST Kather et al. (2016)), Magnetic Resonance Imaging (BTMRI Nickparvar (2021)), Optical Coherence Tomography (OCTMNIST Kermany et al. (2018)), Ultrasound (BUSI Al-Dhabyani et al. (2020)), and X-Ray (COVID-QU-Ex Tahir et al. (2021) and KneeXray Chen (2018)). The details are listed in Table 3. We utilized the classification accuracy as the metric to evaluate our TAME and other competing methods.

### 4.2 IMPLEMENTATION DETAILS

We employed the pre-trained BiomedCLIP Zhang et al. (2023) with a ViT-B/16 backbone as the victim model and reported the averaged results calculated across three trials (*i.e.*, setting the random seed to 0, 1, and 2). The attack budget $\epsilon_p$ was set to 1/255 to guarantee imperceptible perturbations, and the update step for each attack method was set to 10. All experiments are conducted under the white-box setting, where the adversary has full access to the victim model. In our TAME, the weak transformation strategy is empirically defined as the combination of $\pm 10°$ random rotation and $90\%$ random cropping. The strong transformation strategy applies a more intensive combination of $\pm 30°$

random rotation and 50% random cropping. Since the defense is operated by the user at test time, there is no need for the adversarial restoration map to be undetectable, allowing a large defense budget. Therefore, we set both the defense budget $\epsilon_a$ and the step-size $\alpha$ to $8/255$.

## 4.3 EXPERIMENTAL RESULTS

Table 1: Zero-shot adversarial robustness (%) of our TAME, the BiomedCLIP baseline, and other competing TAD methods on 11 medical datasets. We report the mean and standard deviation calculated across three trials. For each dataset, the highest performance under the Clean, PGD, C&W, and AutoAttack (AA) settings is highlighted in **red**, **blue**, **green**, and **purple**, respectively.

| Dataset | Attack | BiomedCLIP | Anti-Adv | HedgeDefense | TTC | R-TPT | TAME |
|---|---|---|---|---|---|---|---|
| **BTMRI** | Clean | 56.79 | $41.62_{\pm0.10}$ | $\mathbf{58.77}_{\pm0.08}$ | $42.07_{\pm0.43}$ | $54.30_{\pm0.34}$ | $54.13_{\pm1.09}$ |
| | PGD | $0.68_{\pm0.07}$ | $9.88_{\pm0.29}$ | $6.37_{\pm0.10}$ | $53.41_{\pm0.08}$ | $48.67_{\pm0.47}$ | $\mathbf{61.21}_{\pm0.33}$ |
| | C&W | $0.68_{\pm0.03}$ | $7.96_{\pm0.35}$ | $7.55_{\pm0.07}$ | $53.37_{\pm0.67}$ | $48.84_{\pm0.22}$ | $\mathbf{61.50}_{\pm0.43}$ |
| | AA | $0.06_{\pm0.00}$ | $8.15_{\pm0.22}$ | $7.65_{\pm0.15}$ | $56.92_{\pm0.87}$ | $50.46_{\pm0.26}$ | $\mathbf{61.25}_{\pm0.19}$ |
| **BUSI** | Clean | 59.75 | $27.54_{\pm0.00}$ | $49.72_{\pm0.72}$ | $40.96_{\pm1.21}$ | $45.48_{\pm1.11}$ | $\mathbf{62.71}_{\pm2.07}$ |
| | PGD | $0.00_{\pm0.00}$ | $8.33_{\pm0.20}$ | $2.83_{\pm0.53}$ | $51.84_{\pm1.60}$ | $34.88_{\pm1.44}$ | $\mathbf{68.08}_{\pm2.45}$ |
| | C&W | $0.00_{\pm0.00}$ | $14.41_{\pm0.35}$ | $5.37_{\pm1.00}$ | $49.15_{\pm1.83}$ | $34.32_{\pm0.92}$ | $\mathbf{70.90}_{\pm0.53}$ |
| | AA | $0.00_{\pm0.00}$ | $8.47_{\pm0.00}$ | $4.38_{\pm0.53}$ | $55.09_{\pm1.51}$ | $37.71_{\pm1.51}$ | $\mathbf{65.54}_{\pm0.20}$ |
| **COVID-QU-Ex** | Clean | 43.82 | $43.80_{\pm0.01}$ | $\mathbf{48.67}_{\pm0.21}$ | $31.50_{\pm0.37}$ | $37.51_{\pm0.15}$ | $36.38_{\pm0.26}$ |
| | PGD | $0.00_{\pm0.00}$ | $0.15_{\pm0.05}$ | $0.62_{\pm0.01}$ | $48.93_{\pm0.28}$ | $25.99_{\pm0.09}$ | $\mathbf{54.41}_{\pm0.22}$ |
| | C&W | $0.00_{\pm0.00}$ | $0.17_{\pm0.05}$ | $0.66_{\pm0.08}$ | $49.30_{\pm0.31}$ | $26.23_{\pm0.13}$ | $\mathbf{53.70}_{\pm0.42}$ |
| | AA | $0.00_{\pm0.00}$ | $0.20_{\pm0.03}$ | $10.03_{\pm0.06}$ | $40.51_{\pm0.40}$ | $31.76_{\pm0.29}$ | $\mathbf{54.00}_{\pm0.48}$ |
| **CTKIDNEY** | Clean | 42.43 | $40.25_{\pm0.01}$ | $42.56_{\pm0.04}$ | $29.73_{\pm0.14}$ | $\mathbf{48.33}_{\pm0.03}$ | $40.36_{\pm0.38}$ |
| | PGD | $0.87_{\pm0.03}$ | $1.31_{\pm0.04}$ | $2.36_{\pm0.12}$ | $26.32_{\pm0.29}$ | $40.98_{\pm0.13}$ | $\mathbf{53.01}_{\pm0.60}$ |
| | C&W | $0.88_{\pm0.02}$ | $2.75_{\pm0.09}$ | $2.89_{\pm0.04}$ | $26.35_{\pm0.39}$ | $41.10_{\pm0.12}$ | $\mathbf{52.02}_{\pm0.60}$ |
| | AA | $0.05_{\pm0.00}$ | $0.68_{\pm0.04}$ | $4.91_{\pm0.12}$ | $32.58_{\pm0.25}$ | $45.23_{\pm0.31}$ | $\mathbf{50.42}_{\pm0.61}$ |
| **DermaMNIST** | Clean | 38.80 | $38.65_{\pm0.00}$ | $37.44_{\pm0.14}$ | $15.69_{\pm0.33}$ | $\mathbf{43.09}_{\pm0.39}$ | $27.95_{\pm0.63}$ |
| | PGD | $0.00_{\pm0.00}$ | $0.07_{\pm0.06}$ | $0.88_{\pm0.02}$ | $\mathbf{40.57}_{\pm0.24}$ | $21.00_{\pm0.27}$ | $40.28_{\pm0.59}$ |
| | C&W | $0.00_{\pm0.00}$ | $0.13_{\pm0.02}$ | $1.02_{\pm0.18}$ | $39.98_{\pm1.40}$ | $20.03_{\pm0.12}$ | $\mathbf{41.30}_{\pm0.60}$ |
| | AA | $0.00_{\pm0.00}$ | $0.30_{\pm0.04}$ | $6.23_{\pm0.04}$ | $33.47_{\pm0.48}$ | $33.44_{\pm0.31}$ | $\mathbf{41.99}_{\pm0.25}$ |
| **Kvasir** | Clean | 54.58 | $42.42_{\pm0.00}$ | $56.09_{\pm0.12}$ | $26.64_{\pm1.22}$ | $\mathbf{56.28}_{\pm0.61}$ | $48.36_{\pm1.00}$ |
| | PGD | $0.00_{\pm0.00}$ | $2.19_{\pm0.40}$ | $0.42_{\pm0.07}$ | $46.16_{\pm0.59}$ | $41.89_{\pm0.45}$ | $\mathbf{59.61}_{\pm0.08}$ |
| | C&W | $0.00_{\pm0.00}$ | $2.75_{\pm0.20}$ | $0.31_{\pm0.14}$ | $43.11_{\pm0.52}$ | $41.42_{\pm0.65}$ | $\mathbf{58.11}_{\pm0.22}$ |
| | AA | $0.00_{\pm0.00}$ | $3.28_{\pm0.04}$ | $4.06_{\pm0.28}$ | $48.19_{\pm0.31}$ | $47.86_{\pm0.34}$ | $\mathbf{63.72}_{\pm0.67}$ |
| **CHMNIST** | Clean | $\mathbf{30.65}$ | $20.39_{\pm0.06}$ | $25.53_{\pm0.14}$ | $25.42_{\pm0.33}$ | $29.94_{\pm0.55}$ | $21.77_{\pm0.71}$ |
| | PGD | $0.00_{\pm0.00}$ | $5.57_{\pm0.30}$ | $0.18_{\pm0.07}$ | $20.15_{\pm0.09}$ | $16.51_{\pm0.37}$ | $\mathbf{25.97}_{\pm0.17}$ |
| | C&W | $0.02_{\pm0.03}$ | $5.30_{\pm0.36}$ | $0.40_{\pm0.06}$ | $19.88_{\pm0.24}$ | $16.58_{\pm0.64}$ | $\mathbf{24.98}_{\pm0.73}$ |
| | AA | $0.00_{\pm0.00}$ | $3.32_{\pm0.11}$ | $3.37_{\pm0.03}$ | $24.96_{\pm0.22}$ | $22.52_{\pm0.38}$ | $\mathbf{30.83}_{\pm0.32}$ |
| **LC25000** | Clean | 50.01 | $48.10_{\pm0.04}$ | $\mathbf{54.14}_{\pm0.07}$ | $32.77_{\pm0.10}$ | $50.12_{\pm0.14}$ | $44.04_{\pm0.23}$ |
| | PGD | $0.01_{\pm0.00}$ | $1.21_{\pm0.03}$ | $0.21_{\pm0.02}$ | $32.21_{\pm0.08}$ | $38.87_{\pm0.05}$ | $\mathbf{55.75}_{\pm0.56}$ |
| | C&W | $0.02_{\pm0.01}$ | $1.48_{\pm0.04}$ | $0.36_{\pm0.03}$ | $30.89_{\pm0.11}$ | $38.43_{\pm0.14}$ | $\mathbf{52.47}_{\pm0.41}$ |
| | AA | $0.01_{\pm0.00}$ | $5.74_{\pm0.01}$ | $8.76_{\pm0.03}$ | $41.19_{\pm0.33}$ | $43.14_{\pm0.05}$ | $\mathbf{54.62}_{\pm0.20}$ |
| **RETINA** | Clean | 26.26 | $26.37_{\pm0.04}$ | $26.10_{\pm0.51}$ | $29.15_{\pm0.62}$ | $\mathbf{32.89}_{\pm0.81}$ | $26.18_{\pm0.30}$ |
| | PGD | $0.00_{\pm0.00}$ | $9.75_{\pm0.55}$ | $2.60_{\pm0.45}$ | $\mathbf{35.54}_{\pm0.82}$ | $20.27_{\pm0.79}$ | $26.13_{\pm0.53}$ |
| | C&W | $0.00_{\pm0.00}$ | $8.89_{\pm0.23}$ | $1.61_{\pm0.10}$ | $\mathbf{33.62}_{\pm0.70}$ | $21.69_{\pm0.84}$ | $26.21_{\pm0.48}$ |
| | AA | $0.00_{\pm0.00}$ | $9.59_{\pm0.41}$ | $8.65_{\pm0.50}$ | $\mathbf{32.68}_{\pm1.11}$ | $29.94_{\pm0.26}$ | $26.68_{\pm0.21}$ |
| **KneeXray** | Clean | 29.47 | $8.86_{\pm0.03}$ | $23.85_{\pm0.05}$ | $24.88_{\pm0.47}$ | $\mathbf{40.84}_{\pm0.31}$ | $38.38_{\pm0.44}$ |
| | PGD | $0.00_{\pm0.00}$ | $0.24_{\pm0.13}$ | $0.22_{\pm0.11}$ | $\mathbf{47.75}_{\pm0.64}$ | $27.70_{\pm0.64}$ | $46.15_{\pm0.33}$ |
| | C&W | $0.00_{\pm0.00}$ | $1.07_{\pm0.11}$ | $1.61_{\pm0.27}$ | $\mathbf{46.94}_{\pm0.39}$ | $28.92_{\pm0.84}$ | $41.08_{\pm0.06}$ |
| | AA | $0.00_{\pm0.00}$ | $3.72_{\pm0.06}$ | $18.74_{\pm0.44}$ | $17.47_{\pm1.07}$ | $35.65_{\pm0.35}$ | $\mathbf{39.61}_{\pm0.26}$ |
| **OCTMNIST** | Clean | 29.90 | $28.80_{\pm0.00}$ | $26.23_{\pm0.05}$ | $29.37_{\pm0.79}$ | $25.40_{\pm0.08}$ | $\mathbf{34.10}_{\pm0.36}$ |
| | PGD | $6.27_{\pm0.68}$ | $8.13_{\pm0.33}$ | $24.83_{\pm0.05}$ | $33.73_{\pm0.28}$ | $25.17_{\pm0.05}$ | $\mathbf{39.40}_{\pm0.29}$ |
| | C&W | $6.37_{\pm0.17}$ | $7.33_{\pm0.33}$ | $25.17_{\pm0.17}$ | $32.57_{\pm0.41}$ | $25.17_{\pm0.05}$ | $\mathbf{39.63}_{\pm0.38}$ |
| | AA | $0.00_{\pm0.00}$ | $0.33_{\pm0.05}$ | $18.30_{\pm0.14}$ | $20.40_{\pm0.59}$ | $25.20_{\pm0.14}$ | $\mathbf{37.10}_{\pm0.57}$ |
| **Average** | Clean | 42.04 | $33.35_{\pm0.01}$ | $40.83_{\pm0.13}$ | $29.84_{\pm0.14}$ | $\mathbf{42.20}_{\pm0.10}$ | $39.49_{\pm0.21}$ |
| | PGD | $0.71_{\pm0.06}$ | $4.26_{\pm0.03}$ | $3.77_{\pm0.04}$ | $39.69_{\pm0.29}$ | $31.08_{\pm0.08}$ | $\mathbf{48.18}_{\pm0.18}$ |
| | C&W | $0.72_{\pm0.01}$ | $4.75_{\pm0.07}$ | $4.27_{\pm0.07}$ | $38.65_{\pm0.24}$ | $31.16_{\pm0.21}$ | $\mathbf{47.45}_{\pm0.14}$ |
| | AA | $0.01_{\pm0.00}$ | $3.98_{\pm0.06}$ | $8.64_{\pm0.05}$ | $36.68_{\pm0.13}$ | $36.63_{\pm0.11}$ | $\mathbf{47.80}_{\pm0.11}$ |

### 4.3.1 COMPARISON WITH OTHER TAD METHODS

We compared our TAME with the BiomedCLIP baseline, two TAD methods designed for the traditional models (Anti-Adv Alfarra et al. (2022) and HedgeDefense Wu et al. (2021)), and two TAD methods tailored for VLMs (TTC Xing et al. (2025) and R-TPT Sheng et al. (2025)). Specifically, we re-implemented all the competing methods using the same baseline and reproduced the results by utilizing their open-source codes. As detailed in Table 1, the results reveal that (1) BiomedCLIP is highly susceptible to adversarial attacks, which devastate its inference capabilities; (2) Anti-Adv and HedgeDefense provide only marginal improvements, underscoring their limited defense ability for

Table 2: Zero-shot adversarial robustness (%) of our TAME, TTC, and R-TPT integrated with three distinct AFT methods: FARE, PMG, and TeCoA. We report the mean and standard deviation calculated across three trials. For each AFT method, the highest performance under the Clean, PGD, C&W, and AutoAttack (AA) settings is highlighted in **red**, **blue**, **green**, and **purple**, respectively.

| Method | Clean | PGD | C&W | AA |
|---|---|---|---|---|
| CLIP (ViT-B/32) | 24.33 | $0.07_{\pm 0.02}$ | $0.13_{\pm 0.01}$ | $0.13_{\pm 0.00}$ |
| FARE | 22.51 | $6.09_{\pm 0.01}$ | $6.02_{\pm 0.01}$ | $5.71_{\pm 0.00}$ |
| FARE + TTC | $23.09_{\pm 0.22}$ | $16.97_{\pm 0.08}$ | $16.40_{\pm 0.03}$ | $22.82_{\pm 0.03}$ |
| FARE + R-TPT | $22.70_{\pm 0.02}$ | $16.77_{\pm 0.03}$ | $17.05_{\pm 0.05}$ | $19.46_{\pm 0.07}$ |
| FARE + TAME (Ours) | $23.43_{\pm 0.10}$ | $32.32_{\pm 0.26}$ | $30.85_{\pm 0.36}$ | $28.98_{\pm 0.11}$ |
| PMG | 22.95 | $12.27_{\pm 0.02}$ | $11.71_{\pm 0.01}$ | $11.65_{\pm 0.02}$ |
| PMG + TTC | $22.48_{\pm 0.05}$ | $16.29_{\pm 0.07}$ | $15.96_{\pm 0.07}$ | $20.12_{\pm 0.13}$ |
| PMG + R-TPT | $20.53_{\pm 0.05}$ | $17.70_{\pm 0.01}$ | $17.51_{\pm 0.04}$ | $19.07_{\pm 0.04}$ |
| PMG + TAME (Ours) | $21.02_{\pm 0.17}$ | $21.40_{\pm 0.14}$ | $21.09_{\pm 0.22}$ | $20.80_{\pm 0.04}$ |
| TeCoA | 22.56 | $11.96_{\pm 0.01}$ | $11.42_{\pm 0.01}$ | $11.49_{\pm 0.01}$ |
| TeCoA + TTC | $22.24_{\pm 0.07}$ | $16.14_{\pm 0.11}$ | $16.02_{\pm 0.13}$ | $20.00_{\pm 0.23}$ |
| TeCoA + R-TPT | $22.84_{\pm 0.05}$ | $19.16_{\pm 0.07}$ | $19.07_{\pm 0.08}$ | $21.34_{\pm 0.10}$ |
| TeCoA + TAME (Ours) | $22.68_{\pm 0.07}$ | $23.52_{\pm 0.05}$ | $23.43_{\pm 0.07}$ | $23.40_{\pm 0.10}$ |

VLMs; and (3) our TAME consistently demonstrates strong adversarial robustness, delivering superior performance in most scenarios and achieving the best overall accuracy across all attack types, while maintaining accuracy on clean images with minor and acceptable degradation. If higher clean accuracy is required, the defense budget can be reduced, as explored in Appendix C.1. Additionally, an intriguing observation is that TAME's overall accuracy under adversarial attacks (48.18% for PGD, 47.45% for C&W, and 47.8% for AutoAttack) surpasses that of BiomedCLIP on clean images (42.04%). This phenomenon indicates a potential risk of label leakage during the attack process. We will discuss it in the Appendix B.

### 4.3.2 EXTENSIBILITY ANALYSIS

To evaluate the extensibility of TAME, TTC, and R-TPT, we integrated each one with various adversarially fine-tuned models. In this experimental setup, we utilized a pre-trained CLIP model with a ViT-B/32 backbone as the base victim model. Due to the challenge of obtaining a fine-tuning dataset that covers all downstream modalities, we implemented three AFT methods (*i.e.*, FARE Schlarmann et al. (2024), PMG Wang et al. (2024a), and TeCoA Mao et al. (2023)) by fine-tuning the CLIP vision encoder on adversarial images from the TinyImageNet dataset. The presence of a significant discrepancy between the adversarial training data and testing data can serve to assess the adaptability of AFT methods in generalizing to unseen testing adversarial images. For conciseness, we only display the average accuracy across 11 datasets in Table 2, and the complete results can be found in Table 8. The results indicate that (1) AFT methods provide only a partial defense against attacks, which can be attributed to their limited adaptation capability when generalized to diverse test data; (2) all TAD methods consistently boost the adversarial robustness of adversarially fine-tuned models, demonstrating the effectiveness of test-time adversarial defense; and (3) our TAME achieves significantly superior robustness enhancements across all adversarial attack types compared to TTC and R-TPT, regardless of the deployed victim model, underscoring its exceptional extensibility.

## 5 CONCLUSION

In this paper, we propose TAME, a novel test-time adversarial defense method designed to improve the zero-shot adversarial robustness of medical vision-language models. By leveraging the semantic fragility of adversarial perturbations, TAME effectively restores model predictions through an adversarial restoration map trained specifically for each test image, requiring only a single update step. To mitigate adverse effects on clean inputs, we further introduce an adaptive weighting mechanism that balances the trade-off between adversarial robustness and clean accuracy, eliminating the need for manual hyperparameter tuning. Extensive experiments across multiple adversarial attacks and 11 medical datasets spanning 9 imaging modalities demonstrate the superiority of our approach, indicating that TAME not only outperforms existing defense strategies but also generalizes effectively to adversarially fine-tuned models. Future work will investigate the extension of this paradigm to a wider range of medical modalities and a more extensive suite of adversarial attacks.

## REPRODUCIBILITY STATEMENT

As recommended, we state the reproducibility of this study here. All 11 medical datasets utilized in this paper are public, and the download links are shown as follows:

- BTMRI: `https://drive.google.com/file/d/1_lJLZRUmczqZqoN-dNq kAzGzmi4ONoU5/view?usp=sharing`

- BUSI: `https://drive.google.com/file/d/1hB5M7wcAUTV9EtiYrijAC oQ36R6VmQaa/view?usp=sharing`

- COVID-QU-Ex: `https://drive.google.com/file/d/1zMLN5q5e_tmH-d eSZQiY4Xq0M1EqCrML/view?usp=sharing`

- CTKidney: `https://drive.google.com/file/d/1PBZ299k--mZL8JU7nh C1Wy8yEmlqmVDh/view?usp=sharing`

- DermaMNIST: `https://drive.google.com/file/d/1Jxd1-DWljunRDZ8f Y80dl5zUMefriQXt/view?usp=sharing`

- Kvasir: `https://drive.google.com/file/d/1T_cqnNIjmGazNeg6gziar vCNWGsFEkRi/view?usp=sharing`

- CHMNIST: `https://drive.google.com/file/d/1tyQiYQmqAGNaY4SCK _8U5vEbbaa1AD-g/view?usp=sharing`

- LC25000: `https://drive.google.com/file/d/1YIu5fqMXgyemisiL1L1 HCvES2nVpCtun/view?usp=sharing`

- RETINA: `https://drive.google.com/file/d/18U-Gc22h5QryomNNzY 4r4Qfrq52yf5EO/view?usp=sharing`

- KneeXray: `https://drive.google.com/file/d/1DBVraYJmxy2UcQ_nGLY vTB2reITOm453/view?usp=sharing`

- OCTMNIST: `https://drive.google.com/file/d/1mYZNWxbPxnnVvcwHQ YybA8gdMzQAoOem/view?usp=sharing`

We followed the data processing pipeline detailed in Koleilat et al. (2025), which is also open-source (`https://github.com/HealthX-Lab/BiomedCoOp/tree/main`). The results of the competing methods are reproduced by using their publicly available source codes, and the corresponding GitHub links are listed below:

- Anti-Adv: `https://github.com/MotasemAlfarra/Combating-Adversa ries-with-Anti-Adversaries`

- HedgeDefense: `https://github.com/burcywu/hedge_defense`

- TTC: `https://github.com/Sxing2/CLIP-Test-time-Counterattack s/tree/main`

- R-TPT: `https://github.com/TomSheng21/R-TPT/tree/main`

- FARE: `https://github.com/chs20/RobustVLM`

- PMG: `https://github.com/serendipity1122/Pre-trained-Model-G uided-Fine-Tuning-for-Zero-Shot-Adversarial-Robustness`

- TeCoA: `https://github.com/cvlab-columbia/ZSRobust4FoundationM odel`

The hyper-parameter configurations of our TAME can be found in Section 4.2, and the code and the computational environment will be available on GitHub.

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

APPENDIX

# A  DATASET DETAILS

Table 3 presents a summary of the 11 medical datasets used for evaluation in this study, encompassing 9 typical biomedical imaging modalities: MRI, ultrasound, X-ray, CT, dermatoscopy, endoscopy, histopathology, CFP, and OCT.

Table 3: Details of 11 datasets across 9 biomedical imaging modalities used in this study.

| Modality | Dataset | Case Number |
|---|---|---|
| Magnetic Resonance Imaging (MRI) | BTMRI | 1717 |
| Ultrasound | BUSI | 236 |
| X-Ray | COVID-QU-Ex | 1656 |
| | KneeXray | 6351 |
| Computerized Tomography (CT) | CTKidney | 3738 |
| Dermatoscopy | DermaMNIST | 2005 |
| Endoscopy | Kvasir | 1200 |
| Histopathology | CHMNIST | 1504 |
| | LC25000 | 7500 |
| Color Fundus Photography (CFP) | RETINA | 1268 |
| Optical Coherence Tomography (OCT) | OCTMNIST | 1000 |

# B  LABEL LEAKAGE BY ATTACKS

To validate the phenomenon of "label leakage", we attacked BiomedCLIP using the PGD method with an attack budget of $1/255$ and a step size of $10$. Specifically, we evaluated three settings: (1) "Chance-level": a chance-level baseline with random guessing; (2) "Random Noise": classification using perturbations initialized from random noise; and (3) "Label As Target": classification using adversarial perturbations generated by the PGD method. For the latter two, We employed a mini-ResNet He et al. (2016) (about $0.3M$ parameters) as a simple classifier, utilized to predict the class labels from the input perturbations. This classifier is trained by an Adam Kingma & Ba (2014) optimizer using a learning rate of $0.001$ for 10 epochs, with an 8:2 train–validation data split. As shown in Figure 7, the "Chance-level" achieves an accuracy of approximately $1/k$, where $k$ denotes the number of categories. The accuracy of "Random Noise" is comparable to this baseline across most datasets, while "Label As Target" exhibits a significantly higher overall accuracy. This finding highlights the potential risk of label leakage via adversarial attacks. We argue that this leakage occurs since adversarial perturbations are optimized along gradient directions that are inherently label-aligned, thereby embedding class-related information at the pixel or feature level. Consequently, a defense approach that learns to recognize and reverse such information could transform adversarial perturbations into signals that are beneficial to the model's performance. This insight suggests that future research should reconsider the supervision strategy of attack methods to mitigate the risk of label leakage.

# C  ADDITIONAL EXPERIMENTAL RESULTS

## C.1  ABLATION STUDY

In this section, we will discuss the effect of the proposed weighting mechanism and analyze the sensitivity of our TAME to the defense budget $\delta_a$ and the step-size $\alpha$. We repeated the experiments on 11 medical datasets using BiomedCLIP as the victim model, and the results are summarized in Table 4. It reveals that (1) the dynamic weight $\omega$ preserves performance on clean images by sacrificing robustness to adversarial images, where the extremely high adversarial robustness intensifies the suspicion of label leakage during adversarial attacks; (2) our TAME is robust to the variation of $\delta_a$ and $\alpha$; and (3) the performance on clean and adversarial images generally exhibits opposite

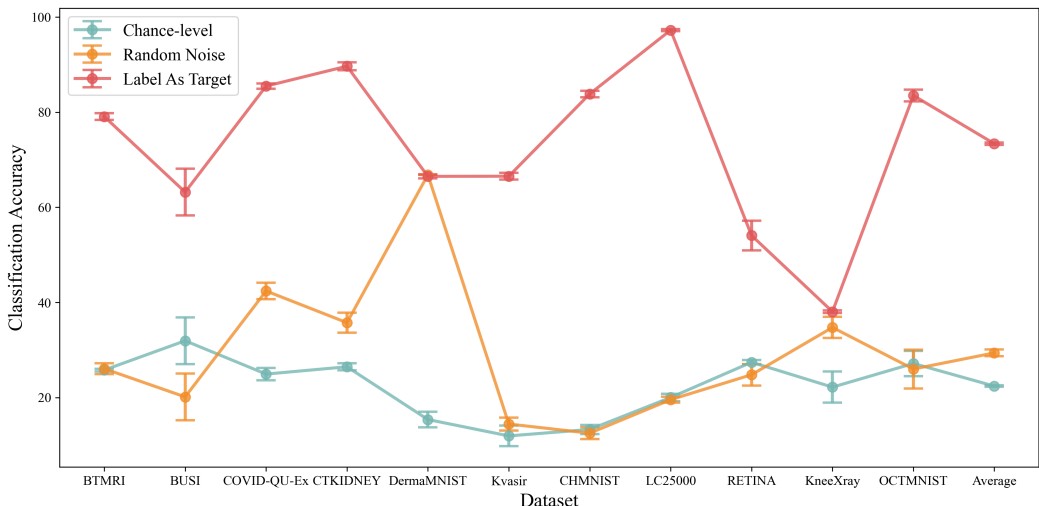

Figure 7: Classification accuracy across 11 datasets under various situations, where the error bars represent standard deviation calculated across 3 trials. Chance-level: the expected performance of making predictions by random guessing. Random Noise: using the perturbations initialized by random noise to train a simple classifier. Label As Target: using the adversarial perturbations produced by PGD to train a simple classifier.

trends as $\delta_a$ and $\alpha$ decrease. The trade-off issue between clean and adversarial robustness will be considered in our future work.

## C.2 LARGER ATTACK BUDGET

We enlarged the attack budget from $1/255$ to $4/255$ to evaluate the effectiveness of our TAME and other compared TAD methods in a more challenging environment. All the experiment configurations of all methods are frozen to avoid additional tuning. Since amplifying the attack budget will not affect the performance on clean images, the results under the clean setting are omitted. We reported the results in Table 6. The results demonstrate that our TAME still remains robust against all attack types even with such a larger attack budget and surpasses other compared methods across most datasets, achieving the best overall accuracy.

## C.3 DEPLOY TO OTHER MEDICAL VLMS

We further discussed the generalizability of the defense methods when deployed to other medical VLMs. The PubMedCLIP Eslami et al. (2023) with a ViT-B/32 backbone is introduced as the victim model, and other experimental configurations remain consistent with those in Table 1. The results displayed in Table 5 reveal that our TAME consistently achieves the strongest overall adversarial robustness against all attack types with an acceptable accuracy on clean images. The label leakage phenomenon can also be clearly observed that the overall accuracies of our TAME against PGD (41.41%), C&W (40.42%), and AutoAttack (36.19%) are much higher than the accuracy of the PubMedCLIP baseline on clean images (27.24%).

## C.4 ADVERSARIAL ROBUSTNESS ON NATURAL IMAGE TASKS

To evaluate the generalizability of our TAME in natural scenes, we followed previous adversarial defense works Xing et al. (2025); Wang et al. (2024a) and conducted experiments on 16 diverse natural image datasets, including four distinct tasks:

- **General Object Recognition:** CIFAR10 Krizhevsky et al. (2009), CIFAR100 Krizhevsky et al. (2009), STL10 Coates et al. (2011), ImageNet Deng et al. (2009), Caltech101 Fei-Fei et al. (2006), and Caltech256 Griffin et al. (2007).

Table 4: Zero-shot adversarial robustness (%) of various variants of our TAME on 11 medical datasets. We report the mean and standard deviation calculated across three trials.

| Dataset | Attack | BiomedCLIP | Ablation study for TAME | | | |
|---|---|---|---|---|---|---|
| | | | w/o $\omega$ | $\epsilon_a = \alpha = 4/255$ | $\epsilon_a = \alpha = 2/255$ | Ours |
| BTMRI | Clean | 56.79 | $38.36_{\pm1.62}$ | $54.07_{\pm1.09}$ | $55.44_{\pm0.98}$ | $54.13_{\pm1.09}$ |
| | PGD | $0.68_{\pm0.07}$ | $76.84_{\pm0.22}$ | $58.71_{\pm0.09}$ | $54.38_{\pm0.06}$ | $61.21_{\pm0.33}$ |
| | C&W | $0.68_{\pm0.03}$ | $76.32_{\pm0.29}$ | $58.40_{\pm0.40}$ | $54.49_{\pm0.48}$ | $61.50_{\pm0.43}$ |
| | AA | $0.06_{\pm0.00}$ | $71.27_{\pm0.62}$ | $60.84_{\pm0.32}$ | $57.39_{\pm0.24}$ | $61.25_{\pm0.19}$ |
| BUSI | Clean | 59.75 | $35.17_{\pm3.51}$ | $61.72_{\pm2.77}$ | $62.57_{\pm2.69}$ | $62.71_{\pm2.07}$ |
| | PGD | $0.00_{\pm0.00}$ | $74.44_{\pm0.87}$ | $62.99_{\pm2.09}$ | $49.01_{\pm2.30}$ | $68.08_{\pm2.45}$ |
| | C&W | $0.00_{\pm0.00}$ | $75.42_{\pm1.51}$ | $65.68_{\pm1.51}$ | $49.01_{\pm2.23}$ | $70.90_{\pm0.53}$ |
| | AA | $0.00_{\pm0.00}$ | $70.48_{\pm1.00}$ | $66.67_{\pm1.78}$ | $58.61_{\pm2.94}$ | $65.54_{\pm0.20}$ |
| COVID-QU-Ex | Clean | 43.82 | $31.72_{\pm0.30}$ | $43.39_{\pm0.30}$ | $44.38_{\pm0.28}$ | $36.38_{\pm0.26}$ |
| | PGD | $0.00_{\pm0.00}$ | $66.47_{\pm0.18}$ | $47.54_{\pm0.09}$ | $38.67_{\pm0.38}$ | $54.41_{\pm0.22}$ |
| | C&W | $0.00_{\pm0.00}$ | $64.99_{\pm0.04}$ | $47.14_{\pm0.11}$ | $38.46_{\pm0.36}$ | $53.70_{\pm0.42}$ |
| | AA | $0.00_{\pm0.00}$ | $60.33_{\pm0.36}$ | $54.66_{\pm0.69}$ | $46.12_{\pm0.45}$ | $54.00_{\pm0.48}$ |
| CTKIDNEY | Clean | 42.43 | $30.48_{\pm0.47}$ | $43.39_{\pm0.51}$ | $45.15_{\pm1.15}$ | $40.36_{\pm0.38}$ |
| | PGD | $0.87_{\pm0.03}$ | $56.08_{\pm0.09}$ | $48.66_{\pm0.32}$ | $43.03_{\pm0.36}$ | $53.01_{\pm0.60}$ |
| | C&W | $0.88_{\pm0.02}$ | $55.80_{\pm0.06}$ | $48.21_{\pm0.73}$ | $42.86_{\pm0.36}$ | $52.02_{\pm0.60}$ |
| | AA | $0.05_{\pm0.00}$ | $53.14_{\pm0.23}$ | $50.16_{\pm0.36}$ | $47.68_{\pm0.28}$ | $50.42_{\pm0.61}$ |
| DermaMNIST | Clean | 38.80 | $25.06_{\pm0.20}$ | $31.60_{\pm0.42}$ | $34.20_{\pm0.83}$ | $27.95_{\pm0.63}$ |
| | PGD | $0.00_{\pm0.00}$ | $43.67_{\pm0.06}$ | $38.63_{\pm0.44}$ | $28.91_{\pm0.14}$ | $40.28_{\pm0.59}$ |
| | C&W | $0.00_{\pm0.00}$ | $45.50_{\pm0.72}$ | $38.14_{\pm0.28}$ | $28.50_{\pm0.47}$ | $41.30_{\pm0.60}$ |
| | AA | $0.00_{\pm0.00}$ | $42.37_{\pm0.21}$ | $45.22_{\pm0.30}$ | $38.57_{\pm0.31}$ | $41.99_{\pm0.25}$ |
| Kvasir | Clean | 54.58 | $27.64_{\pm1.46}$ | $48.94_{\pm1.29}$ | $50.61_{\pm1.09}$ | $48.36_{\pm1.00}$ |
| | PGD | $0.00_{\pm0.00}$ | $65.72_{\pm0.45}$ | $53.72_{\pm0.14}$ | $46.47_{\pm0.44}$ | $59.61_{\pm0.08}$ |
| | C&W | $0.00_{\pm0.00}$ | $63.83_{\pm0.54}$ | $52.30_{\pm0.34}$ | $46.45_{\pm0.17}$ | $58.11_{\pm0.22}$ |
| | AA | $0.00_{\pm0.00}$ | $66.92_{\pm0.71}$ | $61.80_{\pm0.32}$ | $56.03_{\pm0.49}$ | $63.72_{\pm0.67}$ |
| CHMNIST | Clean | 30.65 | $18.15_{\pm0.79}$ | $29.77_{\pm0.42}$ | $30.65_{\pm0.30}$ | $21.77_{\pm0.71}$ |
| | PGD | $0.00_{\pm0.00}$ | $28.81_{\pm0.11}$ | $30.52_{\pm0.80}$ | $26.02_{\pm0.58}$ | $25.97_{\pm0.17}$ |
| | C&W | $0.02_{\pm0.03}$ | $28.57_{\pm0.32}$ | $30.96_{\pm0.36}$ | $25.73_{\pm0.74}$ | $24.98_{\pm0.73}$ |
| | AA | $0.00_{\pm0.00}$ | $31.61_{\pm0.26}$ | $37.66_{\pm1.00}$ | $34.35_{\pm0.28}$ | $30.83_{\pm0.32}$ |
| LC25000 | Clean | 50.01 | $36.87_{\pm0.38}$ | $46.71_{\pm0.27}$ | $48.94_{\pm0.02}$ | $44.04_{\pm0.23}$ |
| | PGD | $0.01_{\pm0.00}$ | $59.96_{\pm0.33}$ | $51.22_{\pm0.45}$ | $42.66_{\pm0.18}$ | $55.75_{\pm0.56}$ |
| | C&W | $0.02_{\pm0.01}$ | $56.36_{\pm0.06}$ | $48.99_{\pm0.96}$ | $41.11_{\pm0.14}$ | $52.47_{\pm0.41}$ |
| | AA | $0.01_{\pm0.00}$ | $54.81_{\pm0.23}$ | $53.48_{\pm0.22}$ | $48.17_{\pm0.22}$ | $54.62_{\pm0.20}$ |
| RETINA | Clean | 26.26 | $27.55_{\pm0.58}$ | $26.10_{\pm0.17}$ | $25.94_{\pm0.82}$ | $26.18_{\pm0.30}$ |
| | PGD | $0.00_{\pm0.00}$ | $35.04_{\pm0.15}$ | $20.22_{\pm0.73}$ | $15.12_{\pm0.38}$ | $26.13_{\pm0.53}$ |
| | C&W | $0.00_{\pm0.00}$ | $35.89_{\pm0.78}$ | $20.24_{\pm0.60}$ | $14.85_{\pm0.30}$ | $26.21_{\pm0.48}$ |
| | AA | $0.00_{\pm0.00}$ | $34.33_{\pm0.38}$ | $27.50_{\pm0.67}$ | $21.85_{\pm0.51}$ | $26.68_{\pm0.21}$ |
| KneeXray | Clean | 29.47 | $37.80_{\pm0.00}$ | $38.33_{\pm0.30}$ | $37.92_{\pm0.54}$ | $38.38_{\pm0.44}$ |
| | PGD | $0.00_{\pm0.00}$ | $50.14_{\pm0.22}$ | $34.66_{\pm0.54}$ | $25.08_{\pm0.34}$ | $46.15_{\pm0.33}$ |
| | C&W | $0.00_{\pm0.00}$ | $44.67_{\pm0.34}$ | $29.09_{\pm0.45}$ | $20.27_{\pm0.95}$ | $41.08_{\pm0.06}$ |
| | AA | $0.00_{\pm0.00}$ | $40.18_{\pm0.16}$ | $39.77_{\pm0.32}$ | $36.49_{\pm0.25}$ | $39.61_{\pm0.26}$ |
| OCTMNIST | Clean | 29.90 | $34.93_{\pm0.25}$ | $33.90_{\pm0.24}$ | $33.47_{\pm0.95}$ | $34.10_{\pm0.36}$ |
| | PGD | $6.27_{\pm0.68}$ | $41.67_{\pm0.68}$ | $39.23_{\pm0.57}$ | $33.30_{\pm0.08}$ | $39.40_{\pm0.29}$ |
| | C&W | $6.37_{\pm0.17}$ | $41.47_{\pm0.42}$ | $39.53_{\pm0.82}$ | $33.90_{\pm0.08}$ | $39.63_{\pm0.38}$ |
| | AA | $0.00_{\pm0.00}$ | $38.30_{\pm0.57}$ | $36.30_{\pm0.92}$ | $34.87_{\pm0.87}$ | $37.10_{\pm0.57}$ |
| Average | Clean | 42.04 | $31.25_{\pm0.25}$ | $41.63_{\pm0.34}$ | $42.66_{\pm0.12}$ | $39.49_{\pm0.21}$ |
| | PGD | $0.71_{\pm0.06}$ | $54.44_{\pm0.06}$ | $44.19_{\pm0.20}$ | $36.60_{\pm0.19}$ | $48.18_{\pm0.18}$ |
| | C&W | $0.72_{\pm0.01}$ | $53.53_{\pm0.14}$ | $43.52_{\pm0.25}$ | $35.97_{\pm0.20}$ | $47.45_{\pm0.14}$ |
| | AA | $0.01_{\pm0.00}$ | $51.25_{\pm0.12}$ | $48.55_{\pm0.29}$ | $43.65_{\pm0.32}$ | $47.80_{\pm0.11}$ |

- **Domain-specific Classification:** FGVCAircraft Maji et al. (2013), EuroSAT Helber et al. (2019), DTD Cimpoi et al. (2014), and PCAM Bejnordi et al. (2017).

- **Fine-grained Recognition:** OxfordPets Parkhi et al. (2012), Flowers102 Nilsback & Zisserman (2008), Food101 Bossard et al. (2014), and StanfordCars Krause et al. (2013).

- **Scene Understanding:** SUN397 Xiao et al. (2010) and Country211 Radford et al. (2021b).

The pre-trained CLIP served as the victim model, and the PGD method was utilized as the adversary. Following Xing et al. (2025), we set the attack budget $\epsilon_p$ and the number of update steps for PGD to 1/255 and 10, respectively. It should be noted that our TAME was deployed directly without any manual tuning. We compared our TAME with four AFT methods (CLIP-FT Xing et al. (2025), TeCoA Mao et al. (2023), PMG Wang et al. (2024a), and FARE Schlarmann et al. (2024)) and four TAD methods (TTE Pérez et al. (2021), Anti-Adv Alfarra et al. (2022), HD Wu et al. (2021), and TTC Xing et al. (2025)). The results shown in Table 7 demonstrate that our TAME achieves superior performance on 9 downstream datasets and the best overall accuracy.

Table 5: Zero-shot adversarial robustness (%) of our TAME, the PubMedCLIP baseline, and other competing TAD methods on 11 medical datasets. We report the mean and standard deviation calculated across three trials. For each dataset, the highest performance under the Clean, PGD, C&W, and AutoAttack (AA) settings is highlighted in **red**, **blue**, **green**, and **purple**, respectively.

| Dataset | Attack | PubMedCLIP | Anti-Adv | HedgeDefense | TTC | R-TPT | TAME |
|---|---|---|---|---|---|---|---|
| BTMRI | Clean | 40.59 | $40.48_{\pm 0.00}$ | $48.13_{\pm 0.11}$ | $40.26_{\pm 0.34}$ | $37.47_{\pm 0.21}$ | $34.75_{\pm 0.46}$ |
| | PGD | $0.37_{\pm 0.03}$ | $3.44_{\pm 0.30}$ | $8.17_{\pm 0.12}$ | $51.66_{\pm 0.34}$ | $27.08_{\pm 0.21}$ | $62.90_{\pm 0.46}$ |
| | C&W | $0.41_{\pm 0.00}$ | $4.39_{\pm 0.19}$ | $8.58_{\pm 0.15}$ | $50.73_{\pm 0.22}$ | $26.91_{\pm 0.17}$ | $58.80_{\pm 0.74}$ |
| | AA | $0.06_{\pm 0.00}$ | $3.94_{\pm 0.06}$ | $18.37_{\pm 0.10}$ | $40.46_{\pm 0.56}$ | $30.77_{\pm 0.12}$ | $48.67_{\pm 0.38}$ |
| BUSI | Clean | 54.66 | $54.66_{\pm 0.00}$ | $55.79_{\pm 0.40}$ | $50.57_{\pm 0.20}$ | $54.80_{\pm 0.20}$ | $43.79_{\pm 3.32}$ |
| | PGD | $0.00_{\pm 0.00}$ | $0.00_{\pm 0.00}$ | $4.10_{\pm 0.20}$ | $53.25_{\pm 0.53}$ | $53.53_{\pm 0.20}$ | $72.32_{\pm 0.72}$ |
| | C&W | $0.00_{\pm 0.00}$ | $0.00_{\pm 0.00}$ | $10.31_{\pm 1.11}$ | $54.38_{\pm 2.30}$ | $52.82_{\pm 0.40}$ | $74.44_{\pm 1.11}$ |
| | AA | $0.00_{\pm 0.00}$ | $3.81_{\pm 1.04}$ | $3.25_{\pm 0.20}$ | $24.72_{\pm 1.06}$ | $54.94_{\pm 0.20}$ | $75.28_{\pm 0.72}$ |
| COVID-QU-Ex | Clean | 6.61 | $7.43_{\pm 0.00}$ | $6.79_{\pm 0.02}$ | $15.78_{\pm 0.25}$ | $6.63_{\pm 0.02}$ | $11.13_{\pm 0.37}$ |
| | PGD | $0.00_{\pm 0.00}$ | $5.83_{\pm 0.16}$ | $0.04_{\pm 0.02}$ | $11.41_{\pm 0.15}$ | $5.67_{\pm 0.01}$ | $17.17_{\pm 0.30}$ |
| | C&W | $0.00_{\pm 0.00}$ | $6.38_{\pm 0.04}$ | $0.09_{\pm 0.01}$ | $11.21_{\pm 0.10}$ | $5.69_{\pm 0.05}$ | $16.35_{\pm 0.20}$ |
| | AA | $0.02_{\pm 0.00}$ | $1.79_{\pm 0.01}$ | $0.32_{\pm 0.03}$ | $12.32_{\pm 0.14}$ | $6.22_{\pm 0.03}$ | $14.25_{\pm 0.24}$ |
| CTKIDNEY | Clean | 22.82 | $22.82_{\pm 0.00}$ | $18.98_{\pm 0.07}$ | $21.62_{\pm 0.19}$ | $23.46_{\pm 0.10}$ | $21.85_{\pm 0.20}$ |
| | PGD | $0.44_{\pm 0.03}$ | $0.44_{\pm 0.03}$ | $1.75_{\pm 0.05}$ | $25.62_{\pm 0.27}$ | $19.73_{\pm 0.16}$ | $37.69_{\pm 0.27}$ |
| | C&W | $0.95_{\pm 0.02}$ | $0.94_{\pm 0.02}$ | $4.11_{\pm 0.15}$ | $28.83_{\pm 0.07}$ | $19.15_{\pm 0.11}$ | $39.41_{\pm 0.39}$ |
| | AA | $0.05_{\pm 0.00}$ | $0.05_{\pm 0.00}$ | $4.00_{\pm 0.10}$ | $20.35_{\pm 0.51}$ | $21.41_{\pm 0.01}$ | $29.05_{\pm 0.38}$ |
| DermaMNIST | Clean | 16.36 | $16.06_{\pm 0.00}$ | $27.48_{\pm 0.12}$ | $20.42_{\pm 0.61}$ | $18.29_{\pm 0.08}$ | $16.23_{\pm 0.39}$ |
| | PGD | $0.00_{\pm 0.00}$ | $0.13_{\pm 0.05}$ | $3.19_{\pm 0.15}$ | $15.79_{\pm 0.22}$ | $14.51_{\pm 0.07}$ | $29.31_{\pm 0.09}$ |
| | C&W | $0.00_{\pm 0.00}$ | $0.23_{\pm 0.06}$ | $4.66_{\pm 0.22}$ | $15.35_{\pm 0.27}$ | $14.00_{\pm 0.16}$ | $26.63_{\pm 0.27}$ |
| | AA | $0.00_{\pm 0.00}$ | $0.37_{\pm 0.02}$ | $15.16_{\pm 0.12}$ | $17.92_{\pm 0.41}$ | $16.68_{\pm 0.17}$ | $24.82_{\pm 0.27}$ |
| Kvasir | Clean | 13.00 | $12.83_{\pm 0.00}$ | $12.92_{\pm 0.12}$ | $13.58_{\pm 0.54}$ | $13.03_{\pm 0.08}$ | $9.81_{\pm 0.32}$ |
| | PGD | $0.00_{\pm 0.00}$ | $0.03_{\pm 0.04}$ | $0.33_{\pm 0.07}$ | $14.97_{\pm 0.42}$ | $12.17_{\pm 0.07}$ | $19.03_{\pm 0.08}$ |
| | C&W | $0.00_{\pm 0.00}$ | $0.19_{\pm 0.04}$ | $0.64_{\pm 0.04}$ | $15.03_{\pm 0.28}$ | $12.17_{\pm 0.07}$ | $18.53_{\pm 0.21}$ |
| | AA | $0.00_{\pm 0.00}$ | $0.00_{\pm 0.00}$ | $0.36_{\pm 0.04}$ | $16.08_{\pm 0.30}$ | $12.86_{\pm 0.17}$ | $16.75_{\pm 0.12}$ |
| CHMNIST | Clean | 20.48 | $19.61_{\pm 0.00}$ | $17.60_{\pm 0.40}$ | $14.23_{\pm 0.14}$ | $22.78_{\pm 0.13}$ | $27.68_{\pm 0.37}$ |
| | PGD | $0.00_{\pm 0.00}$ | $0.00_{\pm 0.00}$ | $0.16_{\pm 0.03}$ | $18.57_{\pm 0.42}$ | $18.48_{\pm 0.42}$ | $34.62_{\pm 0.27}$ |
| | C&W | $0.00_{\pm 0.00}$ | $0.07_{\pm 0.00}$ | $0.24_{\pm 0.14}$ | $18.53_{\pm 0.13}$ | $18.28_{\pm 0.30}$ | $33.42_{\pm 0.27}$ |
| | AA | $0.00_{\pm 0.00}$ | $0.00_{\pm 0.00}$ | $1.64_{\pm 0.21}$ | $17.42_{\pm 0.00}$ | $21.23_{\pm 0.31}$ | $33.56_{\pm 0.36}$ |
| LC25000 | Clean | 20.71 | $20.71_{\pm 0.00}$ | $20.59_{\pm 0.05}$ | $20.26_{\pm 0.05}$ | $19.92_{\pm 0.01}$ | $22.63_{\pm 0.37}$ |
| | PGD | $1.07_{\pm 0.03}$ | $1.09_{\pm 0.04}$ | $4.17_{\pm 0.09}$ | $16.37_{\pm 0.08}$ | $19.43_{\pm 0.02}$ | $38.40_{\pm 0.23}$ |
| | C&W | $1.62_{\pm 0.03}$ | $1.65_{\pm 0.02}$ | $4.49_{\pm 0.04}$ | $16.09_{\pm 0.02}$ | $19.37_{\pm 0.03}$ | $38.09_{\pm 0.20}$ |
| | AA | $0.20_{\pm 0.00}$ | $0.24_{\pm 0.01}$ | $2.39_{\pm 0.04}$ | $19.71_{\pm 0.05}$ | $19.62_{\pm 0.03}$ | $39.85_{\pm 0.07}$ |
| RETINA | Clean | 28.31 | $28.39_{\pm 0.00}$ | $28.86_{\pm 0.17}$ | $24.53_{\pm 0.23}$ | $29.15_{\pm 0.10}$ | $28.79_{\pm 0.74}$ |
| | PGD | $0.00_{\pm 0.00}$ | $0.39_{\pm 0.06}$ | $1.52_{\pm 0.10}$ | $39.04_{\pm 1.01}$ | $20.48_{\pm 0.42}$ | $50.50_{\pm 0.16}$ |
| | C&W | $0.00_{\pm 0.00}$ | $1.16_{\pm 0.15}$ | $1.81_{\pm 0.17}$ | $38.83_{\pm 0.83}$ | $20.45_{\pm 0.13}$ | $50.34_{\pm 0.26}$ |
| | AA | $0.00_{\pm 0.00}$ | $2.68_{\pm 0.17}$ | $5.23_{\pm 0.16}$ | $24.16_{\pm 0.48}$ | $25.47_{\pm 0.19}$ | $33.41_{\pm 0.21}$ |
| KneeXray | Clean | 38.65 | $38.89_{\pm 0.00}$ | $38.51_{\pm 0.03}$ | $34.56_{\pm 0.22}$ | $38.65_{\pm 0.00}$ | $35.41_{\pm 0.45}$ |
| | PGD | $0.00_{\pm 0.00}$ | $0.18_{\pm 0.09}$ | $0.56_{\pm 0.17}$ | $44.44_{\pm 0.35}$ | $28.84_{\pm 0.16}$ | $52.17_{\pm 0.44}$ |
| | C&W | $0.00_{\pm 0.00}$ | $0.48_{\pm 0.05}$ | $0.72_{\pm 0.17}$ | $43.84_{\pm 0.75}$ | $28.95_{\pm 0.17}$ | $51.17_{\pm 0.56}$ |
| | AA | $0.00_{\pm 0.00}$ | $0.30_{\pm 0.00}$ | $1.87_{\pm 0.18}$ | $11.35_{\pm 0.45}$ | $36.11_{\pm 0.23}$ | $43.32_{\pm 0.38}$ |
| OCTMNIST | Clean | 37.50 | $30.00_{\pm 0.00}$ | $39.10_{\pm 0.65}$ | $29.73_{\pm 0.57}$ | $34.47_{\pm 0.09}$ | $27.97_{\pm 1.43}$ |
| | PGD | $0.00_{\pm 0.00}$ | $20.60_{\pm 0.71}$ | $4.43_{\pm 0.33}$ | $43.63_{\pm 0.09}$ | $24.80_{\pm 0.75}$ | $41.40_{\pm 0.54}$ |
| | C&W | $0.00_{\pm 0.00}$ | $21.90_{\pm 0.42}$ | $4.63_{\pm 0.31}$ | $43.80_{\pm 0.14}$ | $24.47_{\pm 0.31}$ | $37.40_{\pm 0.64}$ |
| | AA | $0.10_{\pm 0.00}$ | $19.80_{\pm 0.00}$ | $10.50_{\pm 0.86}$ | $48.30_{\pm 1.63}$ | $28.73_{\pm 0.39}$ | $39.13_{\pm 0.41}$ |
| Average | Clean | 27.24 | $26.54_{\pm 0.00}$ | $28.61_{\pm 0.05}$ | $25.96_{\pm 0.06}$ | $27.15_{\pm 0.03}$ | $25.46_{\pm 0.43}$ |
| | PGD | $0.17_{\pm 0.00}$ | $2.92_{\pm 0.03}$ | $2.58_{\pm 0.03}$ | $30.43_{\pm 0.19}$ | $22.25_{\pm 0.12}$ | $41.41_{\pm 0.11}$ |
| | C&W | $0.27_{\pm 0.00}$ | $3.40_{\pm 0.04}$ | $3.66_{\pm 0.12}$ | $30.60_{\pm 0.34}$ | $22.02_{\pm 0.03}$ | $40.42_{\pm 0.14}$ |
| | AA | $0.04_{\pm 0.00}$ | $3.00_{\pm 0.10}$ | $5.74_{\pm 0.07}$ | $22.98_{\pm 0.26}$ | $24.91_{\pm 0.05}$ | $36.19_{\pm 0.10}$ |

# D COMPLETE RESULTS

Here, we display the complete results for Figure 4, Figure 5, and Table 2 in Figure 8, Figure 9, and Table 8, respectively. As demonstrated in Figure 8, the semantic fragility of adversarial perturbations is observable universally across all 11 datasets, as evidenced by high KL divergence under weak transformations, particularly random cropping and random rotation. Additionally, it can be found that the KL divergence of clean images increases at a markedly higher rate with magnitude than that of their adversarial counterparts on most datasets. This provides powerful evidence for the design of our dynamic weighting mechanism. Figure 9 reveals that both random rotation and random cropping yield a higher robustness ratio across all datasets. This can be attributed to that these two transformations alter the values and/or positions of most pixels in the image and are common in the model training process, thereby leading to low/high robustness on adversarial/clean images. The results in Table 8 indicate that our TAME method boosts the performance of three adversarially fine-tuned models obtained by distinct AFT methods, achieving superior results on most datasets and the highest overall accuracy against all attack types. Additionally, an important finding is the absence

Table 6: Zero-shot adversarial robustness (%) of our TAME, the BiomedCLIP baseline, and other competing TAD methods on 11 medical datasets with a larger attack budget of $4/255$. We report the mean and standard deviation calculated across three trials. For each dataset, the highest performance under the PGD, C&W, and AutoAttack (AA) settings is highlighted in **blue**, **green**, and **purple**, respectively.

| Dataset | Attack | BiomedCLIP | Anti-Adv | HedgeDefense | TTC | R-TPT | TAME (Ours) |
|---|---|---|---|---|---|---|---|
| BTMRI | PGD | $0.02_{\pm0.03}$ | $4.33_{\pm0.24}$ | $0.04_{\pm0.03}$ | $16.02_{\pm0.50}$ | $41.37_{\pm0.27}$ | $38.38_{\pm0.41}$ |
| | C&W | $0.02_{\pm0.03}$ | $3.65_{\pm0.10}$ | $0.04_{\pm0.03}$ | $16.93_{\pm0.47}$ | $42.30_{\pm0.34}$ | $38.73_{\pm0.16}$ |
| | AA | $0.02_{\pm0.03}$ | $7.20_{\pm0.29}$ | $4.85_{\pm0.29}$ | $19.92_{\pm0.69}$ | $48.71_{\pm0.43}$ | $43.37_{\pm0.34}$ |
| BUSI | PGD | $0.00_{\pm0.00}$ | $0.14_{\pm0.20}$ | $0.00_{\pm0.00}$ | $12.15_{\pm0.40}$ | $26.98_{\pm1.06}$ | $25.14_{\pm1.63}$ |
| | C&W | $0.00_{\pm0.00}$ | $0.85_{\pm0.69}$ | $0.00_{\pm0.00}$ | $14.12_{\pm0.40}$ | $25.57_{\pm1.11}$ | $26.41_{\pm1.31}$ |
| | AA | $0.00_{\pm0.00}$ | $3.81_{\pm0.00}$ | $2.97_{\pm0.35}$ | $22.74_{\pm1.56}$ | $39.83_{\pm1.25}$ | $41.38_{\pm3.14}$ |
| COVID-QU-Ex | PGD | $0.00_{\pm0.00}$ | $0.01_{\pm0.01}$ | $0.00_{\pm0.00}$ | $17.05_{\pm0.38}$ | $12.03_{\pm0.03}$ | $27.66_{\pm0.41}$ |
| | C&W | $0.00_{\pm0.00}$ | $0.00_{\pm0.00}$ | $0.00_{\pm0.00}$ | $17.17_{\pm0.67}$ | $13.03_{\pm0.13}$ | $27.84_{\pm0.17}$ |
| | AA | $0.00_{\pm0.00}$ | $0.18_{\pm0.02}$ | $9.77_{\pm0.07}$ | $18.04_{\pm0.07}$ | $19.94_{\pm0.14}$ | $34.06_{\pm0.11}$ |
| CTKIDNEY | PGD | $0.00_{\pm0.00}$ | $0.14_{\pm0.07}$ | $0.00_{\pm0.00}$ | $6.69_{\pm0.23}$ | $29.23_{\pm0.13}$ | $39.42_{\pm0.59}$ |
| | C&W | $0.00_{\pm0.00}$ | $0.60_{\pm0.07}$ | $0.00_{\pm0.00}$ | $6.20_{\pm0.09}$ | $29.28_{\pm0.08}$ | $39.18_{\pm0.01}$ |
| | AA | $0.00_{\pm0.00}$ | $0.35_{\pm0.08}$ | $3.98_{\pm0.09}$ | $7.31_{\pm0.68}$ | $38.29_{\pm0.56}$ | $39.59_{\pm0.25}$ |
| DermaMNIST | PGD | $0.00_{\pm0.00}$ | $0.10_{\pm0.07}$ | $0.00_{\pm0.00}$ | $3.76_{\pm0.15}$ | $5.60_{\pm0.24}$ | $12.67_{\pm0.43}$ |
| | C&W | $0.00_{\pm0.00}$ | $0.15_{\pm0.07}$ | $0.02_{\pm0.02}$ | $4.22_{\pm0.20}$ | $5.50_{\pm0.22}$ | $13.40_{\pm0.20}$ |
| | AA | $0.00_{\pm0.00}$ | $0.25_{\pm0.04}$ | $5.84_{\pm0.23}$ | $8.21_{\pm0.22}$ | $25.09_{\pm0.40}$ | $21.45_{\pm0.99}$ |
| Kvasir | PGD | $0.00_{\pm0.00}$ | $1.22_{\pm0.26}$ | $0.00_{\pm0.00}$ | $7.08_{\pm0.72}$ | $25.22_{\pm0.21}$ | $28.31_{\pm0.40}$ |
| | C&W | $0.00_{\pm0.00}$ | $1.61_{\pm0.17}$ | $0.00_{\pm0.00}$ | $6.25_{\pm0.20}$ | $25.42_{\pm0.30}$ | $27.00_{\pm0.47}$ |
| | AA | $0.00_{\pm0.00}$ | $2.69_{\pm0.24}$ | $4.31_{\pm0.08}$ | $15.17_{\pm0.18}$ | $42.92_{\pm0.36}$ | $43.31_{\pm0.55}$ |
| CHMNIST | PGD | $0.00_{\pm0.00}$ | $4.14_{\pm0.17}$ | $0.00_{\pm0.00}$ | $1.91_{\pm0.13}$ | $8.53_{\pm0.22}$ | $6.78_{\pm0.73}$ |
| | C&W | $0.00_{\pm0.00}$ | $4.19_{\pm0.09}$ | $0.04_{\pm0.03}$ | $2.08_{\pm0.30}$ | $8.22_{\pm0.58}$ | $7.58_{\pm0.29}$ |
| | AA | $0.00_{\pm0.00}$ | $2.44_{\pm0.19}$ | $3.75_{\pm0.19}$ | $11.95_{\pm0.49}$ | $19.44_{\pm0.58}$ | $17.95_{\pm0.30}$ |
| LC25000 | PGD | $0.00_{\pm0.00}$ | $0.01_{\pm0.00}$ | $0.00_{\pm0.00}$ | $2.11_{\pm0.13}$ | $27.40_{\pm0.20}$ | $33.71_{\pm0.27}$ |
| | C&W | $0.00_{\pm0.00}$ | $0.03_{\pm0.02}$ | $0.00_{\pm0.00}$ | $2.16_{\pm0.14}$ | $26.97_{\pm0.16}$ | $33.75_{\pm0.38}$ |
| | AA | $0.00_{\pm0.00}$ | $4.84_{\pm0.02}$ | $8.73_{\pm0.04}$ | $12.45_{\pm0.11}$ | $39.76_{\pm0.17}$ | $43.20_{\pm0.21}$ |
| RETINA | PGD | $0.00_{\pm0.00}$ | $4.60_{\pm0.50}$ | $0.08_{\pm0.06}$ | $11.86_{\pm0.29}$ | $3.68_{\pm0.39}$ | $10.33_{\pm0.29}$ |
| | C&W | $0.00_{\pm0.00}$ | $4.57_{\pm0.23}$ | $0.08_{\pm0.06}$ | $11.04_{\pm0.39}$ | $8.91_{\pm0.06}$ | $9.94_{\pm0.17}$ |
| | AA | $0.00_{\pm0.00}$ | $8.44_{\pm0.06}$ | $6.97_{\pm0.23}$ | $20.11_{\pm0.87}$ | $28.71_{\pm0.46}$ | $17.64_{\pm0.84}$ |
| KneeXray | PGD | $0.00_{\pm0.00}$ | $0.00_{\pm0.00}$ | $0.00_{\pm0.00}$ | $16.20_{\pm0.76}$ | $6.08_{\pm0.25}$ | $34.88_{\pm0.25}$ |
| | C&W | $0.00_{\pm0.00}$ | $0.08_{\pm0.06}$ | $0.00_{\pm0.00}$ | $16.06_{\pm0.64}$ | $32.45_{\pm0.47}$ | $33.39_{\pm0.82}$ |
| | AA | $0.00_{\pm0.00}$ | $3.70_{\pm0.03}$ | $18.78_{\pm0.65}$ | $19.97_{\pm0.84}$ | $34.10_{\pm0.30}$ | $38.45_{\pm0.16}$ |
| OCTMNIST | PGD | $0.00_{\pm0.00}$ | $0.60_{\pm0.28}$ | $0.43_{\pm0.29}$ | $15.67_{\pm1.01}$ | $25.20_{\pm0.00}$ | $30.20_{\pm0.22}$ |
| | C&W | $0.07_{\pm0.05}$ | $0.60_{\pm0.08}$ | $0.43_{\pm0.21}$ | $14.73_{\pm0.45}$ | $25.20_{\pm0.08}$ | $30.50_{\pm0.75}$ |
| | AA | $0.00_{\pm0.00}$ | $0.40_{\pm0.00}$ | $0.20_{\pm0.22}$ | $23.13_{\pm0.39}$ | $25.13_{\pm0.05}$ | $33.03_{\pm0.37}$ |
| Average | PGD | $0.00_{\pm0.00}$ | $1.39_{\pm0.03}$ | $0.05_{\pm0.02}$ | $10.04_{\pm0.13}$ | $19.21_{\pm0.18}$ | $26.14_{\pm0.16}$ |
| | C&W | $0.01_{\pm0.01}$ | $1.48_{\pm0.07}$ | $0.06_{\pm0.02}$ | $10.09_{\pm0.20}$ | $22.08_{\pm0.00}$ | $26.16_{\pm0.32}$ |
| | AA | $0.00_{\pm0.00}$ | $3.12_{\pm0.02}$ | $6.38_{\pm0.10}$ | $16.27_{\pm0.28}$ | $32.90_{\pm0.04}$ | $33.95_{\pm0.29}$ |

Table 7: Zero-shot adversarial robustness (%) of our TAME, the CLIP baseline, and other competing AFT and TAD methods under the PGD attack on 16 datasets. We report the mean and standard deviation calculated across three trials. The results marked by ‡ are inherited from Xing et al. (2025). The best and second-best results in each row are highlighted in **bold** and underline, respectively.

| Dataset | CLIP‡ | Adversarial Fine-tuning (AFT) | | | | Test-time Adversarial Defense (TAD) | | | | |
|---|---|---|---|---|---|---|---|---|---|---|
| | | CLIP-FT‡ | TeCoA‡ | PMG‡ | FARE‡ | TTE‡ | Anti-Adv‡ | HD‡ | TTC‡ | TAME (Ours) |
| CIFAR10 | 0.74 | 3.34 | 33.61 | 40.66 | 19.65 | $41.35_{\pm6.14}$ | $12.39_{\pm0.07}$ | $17.22_{\pm0.45}$ | $28.75_{\pm0.18}$ | $63.20_{\pm0.15}$ |
| CIFAR100 | 0.26 | 0.90 | 18.95 | 22.52 | 11.40 | $20.06_{\pm4.03}$ | $5.73_{\pm0.04}$ | $3.86_{\pm0.10}$ | $14.31_{\pm0.25}$ | $27.64_{\pm0.43}$ |
| STL10 | 11.0 | 12.73 | 70.08 | 73.08 | 59.06 | $78.48_{\pm3.83}$ | $37.42_{\pm0.40}$ | $39.02_{\pm0.30}$ | $76.70_{\pm0.23}$ | $89.23_{\pm0.24}$ |
| ImageNet | 1.15 | 0.93 | 18.89 | 21.43 | 14.00 | $31.01_{\pm4.40}$ | $8.67_{\pm0.05}$ | $6.63_{\pm0.05}$ | $38.41_{\pm0.07}$ | $4.71_{\pm0.05}$ |
| Caltech101 | 14.67 | 14.21 | 55.51 | 61.08 | 50.74 | $67.56_{\pm3.88}$ | $34.81_{\pm0.16}$ | $31.53_{\pm0.22}$ | $65.78_{\pm0.07}$ | $66.83_{\pm0.32}$ |
| Caltech256 | 8.47 | 6.76 | 43.19 | 45.91 | 38.79 | $60.09_{\pm4.03}$ | $25.36_{\pm0.17}$ | $23.48_{\pm0.10}$ | $60.11_{\pm0.04}$ | $48.99_{\pm0.06}$ |
| OxfordPets | 1.04 | 2.10 | 38.35 | 41.18 | 31.07 | $50.33_{\pm7.30}$ | $20.42_{\pm0.22}$ | $12.04_{\pm0.16}$ | $57.87_{\pm0.15}$ | $83.57_{\pm0.40}$ |
| Flowers102 | 1.14 | 0.54 | 21.94 | 23.43 | 17.14 | $35.88_{\pm4.72}$ | $7.16_{\pm0.41}$ | $7.29_{\pm0.06}$ | $39.14_{\pm0.28}$ | $51.48_{\pm0.11}$ |
| FGVCAircraft | 0.00 | 0.00 | 2.49 | 2.22 | 1.35 | $6.23_{\pm1.37}$ | $1.27_{\pm0.07}$ | $1.26_{\pm0.07}$ | $13.77_{\pm0.38}$ | $13.96_{\pm0.23}$ |
| StanfordCars | 0.02 | 0.06 | 8.76 | 11.65 | 6.75 | $22.36_{\pm4.17}$ | $4.40_{\pm0.30}$ | $2.71_{\pm0.09}$ | $33.01_{\pm0.07}$ | $30.46_{\pm0.14}$ |
| SUN397 | 1.14 | 0.94 | 19.39 | 22.58 | 14.91 | $30.79_{\pm4.43}$ | $8.05_{\pm0.04}$ | $6.40_{\pm0.06}$ | $41.52_{\pm0.04}$ | $12.74_{\pm0.08}$ |
| Country211 | 0.04 | 0.03 | 1.78 | 2.12 | 0.85 | $3.05_{\pm0.89}$ | $0.67_{\pm0.05}$ | $0.47_{\pm0.02}$ | $7.09_{\pm0.04}$ | $5.80_{\pm0.03}$ |
| Food101 | 0.70 | 0.42 | 13.90 | 18.57 | 11.65 | $43.94_{\pm6.97}$ | $13.12_{\pm0.16}$ | $8.03_{\pm0.11}$ | $57.84_{\pm0.15}$ | $67.22_{\pm0.23}$ |
| EuroSAT | 0.03 | 0.04 | 11.96 | 12.60 | 10.67 | $6.91_{\pm2.13}$ | $2.15_{\pm0.04}$ | $4.57_{\pm0.09}$ | $12.19_{\pm0.24}$ | $28.40_{\pm0.24}$ |
| DTD | 2.98 | 2.39 | 17.61 | 14.95 | 15.64 | $23.90_{\pm2.34}$ | $5.62_{\pm0.07}$ | $11.63_{\pm0.17}$ | $27.32_{\pm0.25}$ | $24.84_{\pm0.26}$ |
| PCAM | 0.08 | 1.11 | 48.24 | 46.18 | 16.23 | $10.62_{\pm3.22}$ | $4.97_{\pm0.12}$ | $44.74_{\pm0.17}$ | $52.85_{\pm0.20}$ | $80.31_{\pm0.32}$ |
| Average | 2.70 | 2.91 | 26.54 | 28.76 | 20.00 | $33.28_{\pm3.98}$ | $12.01_{\pm0.04}$ | $13.81_{\pm0.06}$ | $39.17_{\pm0.02}$ | $43.71_{\pm0.13}$ |

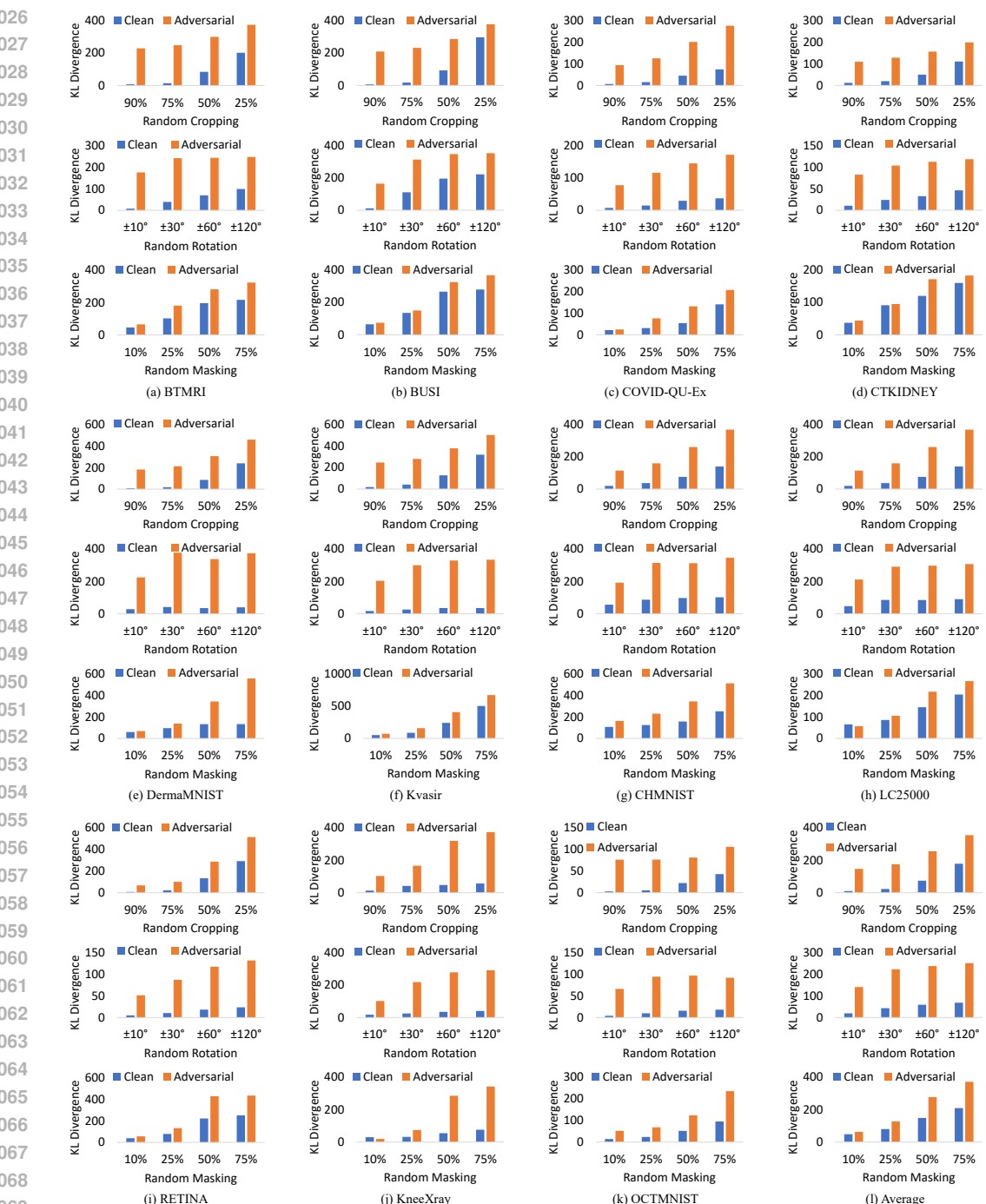

Figure 8: The KL divergence between BiomedCLIP's predictions before and after applying transformations across all 11 datasets with various modalities.

of label leakage when PMG is employed as the AFT method, which suggests that the occurrence of this phenomenon depends on the specific victim model. Note that TeCoA fails on the BUSI dataset, classifying all samples into the same category with high confidence. Consequently, applying other attack or defense strategies yields identical results.

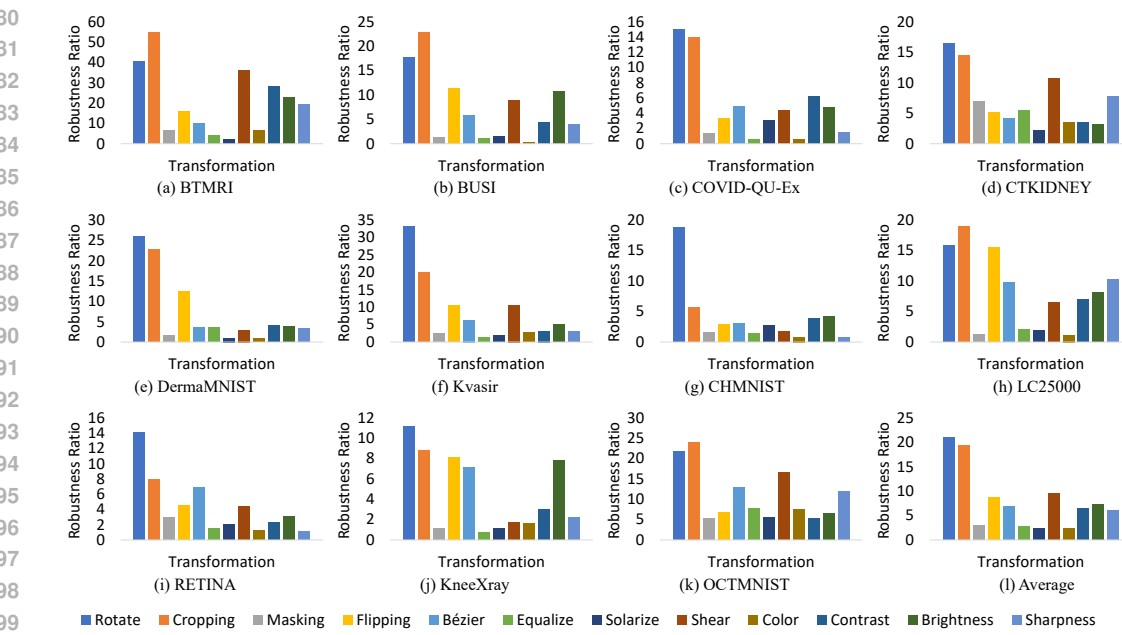

Figure 9: The robustness ratio of 12 transformation strategies across all 11 datasets.

# E  THE USE OF LARGE LANGUAGE MODELS

In this paper, we employed DeepSeek-V3 and GPT-5 as assist tools to polish writing and identify potential grammar or spelling errors.

Table 8: Zero-shot adversarial robustness (%) of our TAME, TTC, and R-TPT integrated with three distinct AFT methods: FARE, PMG, and TeCoA. We report the mean and standard deviation calculated across three trials. For each AFT method, the best results under Clean, PGD, C&W, and AutoAttack (AA) settings are highlighted in **red**, **blue**, **green**, and **purple**, respectively.

| Method | Attack | BTMRI | BUSI | COVID-QU-Ex | CTKIDNEY | DermaMNIST | Kvasir | CHMNIST | LC25000 | RETINA | KneeXray | OCTMNIST | Average |
|---|---|---|---|---|---|---|---|---|---|---|---|---|---|
| CLIP (ViT-B/32) | Clean | 24.64 | 38.56 | 6.36 | 30.71 | 28.83 | 17.08 | 24.87 | 29.76 | 26.50 | 14.49 | 25.80 | 24.33 |
| | PGD | $0.02_{\pm0.03}$ | $0.00_{\pm0.00}$ | $0.19_{\pm0.00}$ | $0.01_{\pm0.01}$ | $0.00_{\pm0.00}$ | $0.00_{\pm0.00}$ | $0.07_{\pm0.05}$ | $0.01_{\pm0.01}$ | $0.45_{\pm0.20}$ | $0.00_{\pm0.00}$ | $0.00_{\pm0.00}$ | $0.07_{\pm0.02}$ |
| | C&W | $0.00_{\pm0.00}$ | $0.00_{\pm0.00}$ | $0.28_{\pm0.02}$ | $0.03_{\pm0.00}$ | $0.00_{\pm0.00}$ | $0.00_{\pm0.00}$ | $0.07_{\pm0.05}$ | $0.01_{\pm0.00}$ | $1.05_{\pm0.21}$ | $0.02_{\pm0.03}$ | $0.00_{\pm0.00}$ | $0.13_{\pm0.01}$ |
| | AA | $0.00_{\pm0.00}$ | $0.42_{\pm0.00}$ | $0.00_{\pm0.00}$ | $0.21_{\pm0.00}$ | $0.05_{\pm0.00}$ | $0.00_{\pm0.00}$ | $0.07_{\pm0.00}$ | $0.32_{\pm0.00}$ | $0.00_{\pm0.00}$ | $0.12_{\pm0.00}$ | $0.20_{\pm0.00}$ | $0.13_{\pm0.00}$ |
| FARE | Clean | 29.76 | 17.37 | 9.35 | 28.06 | 22.19 | 15.08 | 23.34 | 23.43 | 26.10 | 33.39 | 19.50 | 22.51 |
| | PGD | $19.74_{\pm0.08}$ | $16.95_{\pm0.00}$ | $0.84_{\pm0.01}$ | $6.20_{\pm0.02}$ | $3.16_{\pm0.02}$ | $0.17_{\pm0.00}$ | $12.59_{\pm0.03}$ | $4.60_{\pm0.01}$ | $2.44_{\pm0.00}$ | $0.12_{\pm0.00}$ | $0.20_{\pm0.00}$ | $6.09_{\pm0.01}$ |
| | C&W | $19.02_{\pm0.11}$ | $16.95_{\pm0.00}$ | $0.79_{\pm0.01}$ | $5.69_{\pm0.02}$ | $3.11_{\pm0.02}$ | $0.25_{\pm0.00}$ | $12.65_{\pm0.03}$ | $4.84_{\pm0.01}$ | $2.37_{\pm0.00}$ | $0.18_{\pm0.00}$ | $0.37_{\pm0.05}$ | $6.02_{\pm0.01}$ |
| | AA | $18.33_{\pm0.05}$ | $16.95_{\pm0.00}$ | $0.80_{\pm0.01}$ | $5.37_{\pm0.02}$ | $2.66_{\pm0.02}$ | $0.17_{\pm0.00}$ | $12.57_{\pm0.00}$ | $3.82_{\pm0.02}$ | $1.86_{\pm0.04}$ | $0.12_{\pm0.00}$ | $0.17_{\pm0.05}$ | $5.71_{\pm0.00}$ |
| + TTC | Clean | $30.40_{\pm0.46}$ | $17.37_{\pm0.00}$ | $15.56_{\pm0.52}$ | $25.28_{\pm0.28}$ | $19.30_{\pm0.31}$ | $15.14_{\pm0.69}$ | $24.49_{\pm0.54}$ | $28.95_{\pm0.50}$ | $25.89_{\pm0.21}$ | $32.27_{\pm0.46}$ | $19.37_{\pm0.42}$ | $23.09_{\pm0.22}$ |
| | PGD | $27.84_{\pm0.26}$ | $17.37_{\pm0.00}$ | $9.06_{\pm0.16}$ | $28.51_{\pm0.05}$ | $10.47_{\pm0.76}$ | $7.17_{\pm0.53}$ | $15.38_{\pm0.21}$ | $18.47_{\pm0.40}$ | $23.73_{\pm0.33}$ | $17.13_{\pm0.20}$ | $11.50_{\pm0.43}$ | $16.97_{\pm0.08}$ |
| | C&W | $27.47_{\pm0.10}$ | $17.37_{\pm0.00}$ | $8.87_{\pm0.21}$ | $28.80_{\pm0.39}$ | $10.11_{\pm0.42}$ | $7.31_{\pm0.28}$ | $14.98_{\pm0.11}$ | $15.55_{\pm0.20}$ | $22.93_{\pm0.27}$ | $16.47_{\pm0.49}$ | $10.53_{\pm0.54}$ | $16.40_{\pm0.03}$ |
| | AA | $31.16_{\pm0.26}$ | $17.23_{\pm0.20}$ | $16.95_{\pm0.03}$ | $28.46_{\pm0.26}$ | $17.49_{\pm0.80}$ | $12.78_{\pm0.69}$ | $22.19_{\pm0.35}$ | $28.29_{\pm0.16}$ | $29.16_{\pm0.32}$ | $28.46_{\pm0.33}$ | $18.83_{\pm1.02}$ | $22.82_{\pm0.03}$ |
| + R-TPT | Clean | $30.81_{\pm0.10}$ | $17.37_{\pm0.00}$ | $15.37_{\pm0.15}$ | $28.03_{\pm0.15}$ | $20.95_{\pm0.11}$ | $16.39_{\pm0.28}$ | $22.38_{\pm0.20}$ | $18.26_{\pm0.17}$ | $26.10_{\pm0.06}$ | $32.91_{\pm0.08}$ | $21.17_{\pm0.41}$ | $22.70_{\pm0.02}$ |
| | PGD | $27.02_{\pm0.09}$ | $17.37_{\pm0.00}$ | $10.17_{\pm0.19}$ | $23.34_{\pm0.12}$ | $12.60_{\pm0.24}$ | $10.81_{\pm0.40}$ | $17.89_{\pm0.11}$ | $12.90_{\pm0.05}$ | $22.53_{\pm0.37}$ | $18.14_{\pm0.07}$ | $11.67_{\pm0.31}$ | $16.77_{\pm0.03}$ |
| | C&W | $27.28_{\pm0.12}$ | $17.37_{\pm0.00}$ | $10.11_{\pm0.12}$ | $23.15_{\pm0.05}$ | $13.12_{\pm0.07}$ | $11.69_{\pm0.10}$ | $17.67_{\pm0.03}$ | $12.63_{\pm0.13}$ | $22.03_{\pm0.21}$ | $21.01_{\pm0.26}$ | $11.47_{\pm0.49}$ | $17.05_{\pm0.05}$ |
| | AA | $28.96_{\pm0.22}$ | $17.37_{\pm0.00}$ | $13.33_{\pm0.05}$ | $25.42_{\pm0.15}$ | $15.96_{\pm0.29}$ | $13.53_{\pm0.04}$ | $20.26_{\pm0.28}$ | $14.38_{\pm0.06}$ | $23.58_{\pm0.23}$ | $25.22_{\pm0.38}$ | $16.07_{\pm0.25}$ | $19.46_{\pm0.07}$ |
| + TAME (Ours) | Clean | $36.48_{\pm0.07}$ | $17.80_{\pm0.35}$ | $12.55_{\pm0.27}$ | $24.99_{\pm1.00}$ | $17.09_{\pm0.14}$ | $16.28_{\pm0.17}$ | $25.56_{\pm0.33}$ | $27.52_{\pm0.54}$ | $21.95_{\pm0.49}$ | $30.25_{\pm0.47}$ | $27.27_{\pm0.47}$ | $23.43_{\pm0.10}$ |
| | PGD | $45.08_{\pm1.09}$ | $21.19_{\pm1.38}$ | $11.97_{\pm0.18}$ | $33.16_{\pm0.75}$ | $24.06_{\pm0.12}$ | $20.39_{\pm0.20}$ | $33.55_{\pm0.08}$ | $43.44_{\pm0.62}$ | $31.81_{\pm0.42}$ | $57.05_{\pm0.13}$ | $33.83_{\pm0.68}$ | $32.32_{\pm0.26}$ |
| | C&W | $44.73_{\pm0.72}$ | $20.48_{\pm1.40}$ | $11.90_{\pm0.29}$ | $33.63_{\pm1.26}$ | $22.46_{\pm0.20}$ | $18.53_{\pm0.32}$ | $31.87_{\pm0.64}$ | $42.38_{\pm0.45}$ | $31.04_{\pm0.49}$ | $52.46_{\pm0.12}$ | $29.90_{\pm1.00}$ | $30.85_{\pm0.36}$ |
| | AA | $41.04_{\pm0.96}$ | $17.80_{\pm0.35}$ | $13.28_{\pm0.14}$ | $29.81_{\pm0.65}$ | $22.46_{\pm0.12}$ | $21.14_{\pm0.69}$ | $29.17_{\pm0.35}$ | $34.49_{\pm0.46}$ | $29.16_{\pm0.64}$ | $50.69_{\pm0.49}$ | $29.70_{\pm1.10}$ | $28.98_{\pm0.11}$ |
| PMG | Clean | 27.84 | 17.80 | 26.97 | 24.88 | 16.91 | 15.08 | 22.54 | 19.72 | 21.06 | 36.35 | 23.30 | 22.95 |
| | PGD | $23.12_{\pm0.05}$ | $17.37_{\pm0.00}$ | $15.11_{\pm0.01}$ | $14.54_{\pm0.07}$ | $7.43_{\pm0.04}$ | $6.19_{\pm0.08}$ | $15.49_{\pm0.00}$ | $14.39_{\pm0.03}$ | $8.65_{\pm0.07}$ | $4.37_{\pm0.06}$ | $8.30_{\pm0.00}$ | $12.27_{\pm0.02}$ |
| | C&W | $22.97_{\pm0.03}$ | $17.37_{\pm0.00}$ | $14.12_{\pm0.05}$ | $13.71_{\pm0.03}$ | $5.37_{\pm0.05}$ | $5.17_{\pm0.00}$ | $15.16_{\pm0.00}$ | $12.71_{\pm0.01}$ | $7.49_{\pm0.00}$ | $6.18_{\pm0.03}$ | $8.60_{\pm0.00}$ | $11.71_{\pm0.01}$ |
| | AA | $23.04_{\pm0.02}$ | $17.37_{\pm0.00}$ | $14.37_{\pm0.05}$ | $14.00_{\pm0.03}$ | $5.67_{\pm0.08}$ | $5.58_{\pm0.07}$ | $14.58_{\pm0.03}$ | $12.93_{\pm0.05}$ | $7.78_{\pm0.10}$ | $4.91_{\pm0.03}$ | $7.90_{\pm0.00}$ | $11.65_{\pm0.02}$ |
| + TTC | Clean | $29.14_{\pm0.52}$ | $18.64_{\pm0.60}$ | $24.45_{\pm0.06}$ | $24.06_{\pm0.14}$ | $16.46_{\pm0.36}$ | $14.83_{\pm0.18}$ | $22.03_{\pm0.23}$ | $20.70_{\pm0.15}$ | $21.58_{\pm0.38}$ | $32.43_{\pm1.44}$ | $23.00_{\pm0.98}$ | $22.48_{\pm0.05}$ |
| | PGD | $24.85_{\pm0.36}$ | $17.37_{\pm0.00}$ | $18.90_{\pm0.04}$ | $23.74_{\pm0.13}$ | $7.93_{\pm0.16}$ | $9.00_{\pm0.12}$ | $17.44_{\pm0.33}$ | $15.97_{\pm0.07}$ | $13.85_{\pm0.32}$ | $17.71_{\pm0.81}$ | $12.47_{\pm0.12}$ | $16.29_{\pm0.07}$ |
| | C&W | $24.71_{\pm0.39}$ | $17.23_{\pm0.20}$ | $18.89_{\pm0.34}$ | $23.13_{\pm0.17}$ | $7.08_{\pm0.19}$ | $8.97_{\pm0.31}$ | $16.93_{\pm0.45}$ | $15.22_{\pm0.08}$ | $13.72_{\pm0.51}$ | $16.85_{\pm0.30}$ | $12.87_{\pm0.53}$ | $15.96_{\pm0.07}$ |
| | AA | $28.17_{\pm0.36}$ | $18.78_{\pm0.20}$ | $22.81_{\pm0.16}$ | $24.72_{\pm0.21}$ | $11.74_{\pm0.08}$ | $11.36_{\pm0.48}$ | $20.37_{\pm0.13}$ | $18.47_{\pm0.13}$ | $17.59_{\pm0.49}$ | $28.02_{\pm0.21}$ | $19.33_{\pm0.52}$ | $20.12_{\pm0.13}$ |
| + R-TPT | Clean | $27.53_{\pm0.03}$ | $17.37_{\pm0.00}$ | $27.38_{\pm0.04}$ | $24.03_{\pm0.09}$ | $13.02_{\pm0.04}$ | $12.95_{\pm0.17}$ | $22.34_{\pm0.24}$ | $17.64_{\pm0.06}$ | $13.85_{\pm0.43}$ | $31.20_{\pm0.20}$ | $18.50_{\pm0.22}$ | $20.53_{\pm0.05}$ |
| | PGD | $25.92_{\pm0.17}$ | $17.37_{\pm0.00}$ | $25.08_{\pm0.12}$ | $22.31_{\pm0.10}$ | $10.49_{\pm0.06}$ | $12.03_{\pm0.22}$ | $20.66_{\pm0.03}$ | $16.13_{\pm0.08}$ | $11.07_{\pm0.04}$ | $21.14_{\pm0.30}$ | $12.57_{\pm0.05}$ | $17.70_{\pm0.01}$ |
| | C&W | $26.15_{\pm0.05}$ | $17.37_{\pm0.00}$ | $24.83_{\pm0.10}$ | $21.97_{\pm0.16}$ | $9.66_{\pm0.09}$ | $12.03_{\pm0.10}$ | $19.88_{\pm0.20}$ | $15.48_{\pm0.03}$ | $10.49_{\pm0.49}$ | $21.35_{\pm0.53}$ | $13.37_{\pm0.19}$ | $17.51_{\pm0.04}$ |
| | AA | $26.98_{\pm0.14}$ | $17.37_{\pm0.00}$ | $25.81_{\pm0.01}$ | $23.47_{\pm0.08}$ | $11.14_{\pm0.09}$ | $12.61_{\pm0.04}$ | $21.03_{\pm0.11}$ | $16.58_{\pm0.04}$ | $12.09_{\pm0.18}$ | $26.69_{\pm0.13}$ | $16.00_{\pm0.37}$ | $19.07_{\pm0.04}$ |
| + TAME (Ours) | Clean | $30.11_{\pm0.41}$ | $17.37_{\pm0.00}$ | $23.84_{\pm0.46}$ | $23.99_{\pm0.41}$ | $12.24_{\pm0.40}$ | $15.33_{\pm0.20}$ | $21.16_{\pm0.68}$ | $17.49_{\pm0.33}$ | $21.06_{\pm0.26}$ | $26.85_{\pm0.91}$ | $21.80_{\pm0.08}$ | $21.02_{\pm0.17}$ |
| | PGD | $35.74_{\pm1.03}$ | $17.51_{\pm0.20}$ | $23.26_{\pm0.24}$ | $21.98_{\pm0.22}$ | $9.90_{\pm0.12}$ | $14.28_{\pm0.69}$ | $20.81_{\pm0.34}$ | $16.80_{\pm0.15}$ | $17.61_{\pm0.29}$ | $38.91_{\pm0.43}$ | $18.63_{\pm0.17}$ | $21.40_{\pm0.14}$ |
| | C&W | $35.31_{\pm0.76}$ | $17.66_{\pm0.20}$ | $23.47_{\pm0.20}$ | $21.75_{\pm0.40}$ | $8.15_{\pm0.12}$ | $14.31_{\pm0.35}$ | $19.15_{\pm0.14}$ | $17.44_{\pm0.27}$ | $17.80_{\pm0.46}$ | $39.29_{\pm0.84}$ | $17.67_{\pm0.87}$ | $21.09_{\pm0.22}$ |
| | AA | $30.07_{\pm0.34}$ | $17.37_{\pm0.00}$ | $23.58_{\pm0.27}$ | $22.37_{\pm0.38}$ | $9.83_{\pm0.14}$ | $14.67_{\pm0.27}$ | $20.17_{\pm0.17}$ | $16.09_{\pm0.19}$ | $18.17_{\pm0.15}$ | $36.41_{\pm0.70}$ | $20.13_{\pm0.26}$ | $20.80_{\pm0.04}$ |
| TeCoA | Clean | 27.78 | 17.37 | 15.98 | 21.91 | 15.46 | 15.33 | 22.74 | 18.80 | 25.39 | 37.68 | 29.70 | 22.56 |
| | PGD | $27.26_{\pm0.03}$ | $17.37_{\pm0.00}$ | $4.25_{\pm0.01}$ | $13.79_{\pm0.03}$ | $7.50_{\pm0.06}$ | $3.75_{\pm0.07}$ | $13.83_{\pm0.00}$ | $12.95_{\pm0.03}$ | $10.88_{\pm0.07}$ | $9.62_{\pm0.03}$ | $10.40_{\pm0.00}$ | $11.96_{\pm0.01}$ |
| | C&W | $27.26_{\pm0.00}$ | $17.37_{\pm0.00}$ | $1.79_{\pm0.02}$ | $13.33_{\pm0.05}$ | $6.03_{\pm0.00}$ | $3.30_{\pm0.04}$ | $13.76_{\pm0.00}$ | $12.69_{\pm0.04}$ | $10.54_{\pm0.04}$ | $9.64_{\pm0.07}$ | $9.87_{\pm0.05}$ | $11.42_{\pm0.01}$ |
| | AA | $27.26_{\pm0.00}$ | $17.37_{\pm0.00}$ | | $13.48_{\pm0.09}$ | $5.81_{\pm0.10}$ | $3.25_{\pm0.07}$ | $13.47_{\pm0.06}$ | $12.79_{\pm0.02}$ | $10.88_{\pm0.07}$ | $9.76_{\pm0.03}$ | $10.33_{\pm0.12}$ | $11.49_{\pm0.01}$ |
| + TTC | Clean | $27.84_{\pm0.07}$ | $17.37_{\pm0.00}$ | $18.51_{\pm0.29}$ | $22.00_{\pm0.07}$ | $14.56_{\pm0.39}$ | $16.61_{\pm0.37}$ | $22.27_{\pm0.05}$ | $18.96_{\pm0.17}$ | $25.05_{\pm0.19}$ | $34.26_{\pm0.16}$ | $27.20_{\pm0.45}$ | $22.24_{\pm0.07}$ |
| | PGD | $27.70_{\pm0.03}$ | $17.37_{\pm0.00}$ | $11.13_{\pm0.18}$ | $18.67_{\pm0.28}$ | $7.46_{\pm0.45}$ | $8.69_{\pm0.14}$ | $16.13_{\pm0.37}$ | $15.09_{\pm0.14}$ | $17.14_{\pm0.36}$ | $22.71_{\pm0.30}$ | $15.43_{\pm0.63}$ | $16.14_{\pm0.11}$ |
| | C&W | $27.64_{\pm0.07}$ | $17.37_{\pm0.00}$ | $11.22_{\pm0.19}$ | $18.58_{\pm0.18}$ | $6.66_{\pm0.27}$ | $8.58_{\pm0.18}$ | $15.65_{\pm0.31}$ | $15.07_{\pm0.08}$ | $17.30_{\pm0.23}$ | $23.01_{\pm0.95}$ | $15.13_{\pm0.29}$ | $16.02_{\pm0.13}$ |
| | AA | $27.72_{\pm0.08}$ | $17.37_{\pm0.00}$ | $16.96_{\pm0.21}$ | $20.88_{\pm0.23}$ | $10.50_{\pm0.13}$ | $13.78_{\pm1.04}$ | $19.77_{\pm0.39}$ | $16.57_{\pm0.03}$ | $22.16_{\pm0.07}$ | $32.31_{\pm1.02}$ | $21.97_{\pm0.45}$ | $20.00_{\pm0.23}$ |
| + R-TPT | Clean | $27.70_{\pm0.03}$ | $17.37_{\pm0.00}$ | $22.63_{\pm0.03}$ | $21.70_{\pm0.08}$ | $12.74_{\pm0.06}$ | $16.44_{\pm0.14}$ | $21.12_{\pm0.31}$ | $18.83_{\pm0.09}$ | $28.16_{\pm0.40}$ | $31.86_{\pm0.30}$ | $32.67_{\pm0.46}$ | $22.84_{\pm0.05}$ |
| | PGD | $27.45_{\pm0.06}$ | $17.37_{\pm0.00}$ | $18.40_{\pm0.09}$ | $17.59_{\pm0.09}$ | $10.32_{\pm0.04}$ | $11.97_{\pm0.14}$ | $18.33_{\pm0.11}$ | $17.67_{\pm0.02}$ | $22.16_{\pm0.07}$ | $22.10_{\pm0.22}$ | $27.37_{\pm0.71}$ | $19.16_{\pm0.07}$ |
| | C&W | $27.47_{\pm0.08}$ | $17.37_{\pm0.00}$ | $18.31_{\pm0.02}$ | $17.84_{\pm0.15}$ | $9.48_{\pm0.04}$ | $12.61_{\pm0.28}$ | $17.86_{\pm0.17}$ | $17.71_{\pm0.04}$ | $21.85_{\pm0.17}$ | $22.32_{\pm0.15}$ | $26.97_{\pm0.40}$ | $19.07_{\pm0.08}$ |
| | AA | $27.51_{\pm0.06}$ | $17.37_{\pm0.00}$ | $21.38_{\pm0.01}$ | $20.44_{\pm0.26}$ | $11.29_{\pm0.16}$ | $14.64_{\pm0.04}$ | $19.75_{\pm0.09}$ | $18.12_{\pm0.03}$ | $25.45_{\pm0.20}$ | $29.01_{\pm0.41}$ | $29.77_{\pm0.48}$ | $21.34_{\pm0.10}$ |
| + TAME (Ours) | Clean | $28.56_{\pm0.49}$ | $17.37_{\pm0.00}$ | $17.70_{\pm0.23}$ | $21.28_{\pm0.71}$ | $13.45_{\pm0.25}$ | $16.97_{\pm0.55}$ | $22.76_{\pm0.35}$ | $18.54_{\pm0.24}$ | $32.05_{\pm0.55}$ | $32.41_{\pm0.40}$ | $28.37_{\pm0.42}$ | $22.68_{\pm0.07}$ |
| | PGD | $30.13_{\pm0.29}$ | $17.37_{\pm0.00}$ | $15.92_{\pm0.18}$ | $19.88_{\pm1.06}$ | $13.45_{\pm0.25}$ | $16.03_{\pm0.22}$ | $21.19_{\pm0.43}$ | $17.96_{\pm0.23}$ | $32.81_{\pm0.40}$ | $45.61_{\pm0.16}$ | $28.37_{\pm0.24}$ | $23.52_{\pm0.05}$ |
| | C&W | $30.09_{\pm0.43}$ | $17.37_{\pm0.00}$ | $15.98_{\pm0.13}$ | $20.21_{\pm1.14}$ | $11.32_{\pm0.33}$ | $15.53_{\pm0.38}$ | $20.86_{\pm0.40}$ | $18.74_{\pm0.37}$ | $32.78_{\pm0.48}$ | $45.88_{\pm0.80}$ | $28.93_{\pm0.86}$ | $23.43_{\pm0.07}$ |
| | AA | $29.00_{\pm0.70}$ | $17.37_{\pm0.00}$ | $16.94_{\pm0.22}$ | $20.84_{\pm0.79}$ | $12.99_{\pm0.12}$ | $17.47_{\pm0.65}$ | $21.52_{\pm0.41}$ | $17.57_{\pm0.12}$ | $30.76_{\pm1.13}$ | $44.26_{\pm0.15}$ | $28.63_{\pm0.61}$ | $23.40_{\pm0.10}$ |

