# OpenReview forum: "The Attack Means Nothing: Test-time Adversarial Defense Improves Zero-shot Adversarial Robustness for Medical Vision-Language Models"
_ICLR.cc/2026/Conference — ICLR 2026 Conference Withdrawn Submission_

### Official Review · Reviewer_82Za · 2025-10-18

**Soundness:** 3
**Presentation:** 2
**Contribution:** 2
**Rating:** 4
**Confidence:** 4

**Summary:**

This paper proposes a test-time adversarial defense method named TAME, aiming to enhance the zero-shot adversarial robustness of medical VLM by exploiting the “semantic vulnerability” of adversarial perturbations. The method optimizes a recovery graph for each sample during testing to improve the consistency of model predictions under weak image transformations. The approach is evaluated on 11 medical datasets.

**Strengths:**

- The paper addresses the critical and highly relevant problem of adversarial robustness in a safety-critical domain.

- The proposed method is simple, plug-and-play, and test-time only, which is highly practical as it avoids costly model retraining.

- The experimental evaluation is extensive, covering a wide range of medical datasets and modalities, which provides strong initial support for the method's generalizability.

**Weaknesses:**

- The paper does not evaluate against an adaptive attack designed specifically to bypass the TAME defense.
- The core technical idea of TAME is a form of consistency regularization, enforcing prediction invariance under transformations. This is a well-established and general concept in semi-supervised learning and robust training literature, but it is framed as a specific solution for "Medical VLMs.
- The TAME algorithm is generic, however, the paper frames its contribution as a defense for "Medical VLMs." This mischaracterizes the work "it is a general method evaluated in a specific domain, not a domain-specific method".
- The paper's explanation for achieving accuracy on attacked images that is higher than on clean images is questionable. A more plausible explanation is that this phenomenon exposes a flaw in the evaluated white-box threat model.
- Typos, such as "distributions" in line 130 where it should be "contributions." Such errors suggest a lack of careful proofreading.
- Use of multiple colors in tables is not a good idea for an academic paper.

**Questions:**

- To address the most critical weakness, could you design and evaluate against an adaptive attack that assumes full knowledge of the TAME pipeline? For example, by differentiating through the one-step optimization of the restoration map $\delta_a$ when crafting the adversarial perturbation $\delta_p$?
-  Can you clarify how TAME differs fundamentally from prior work on consistency regularization? What is the specific, novel contribution of applying this concept via an optimizable input map at test-time compared to existing approaches?
- Why was the scope of the work framed as being specific to "Medical VLMs" when the method itself is generic?
- Could you provide a sensitivity analysis for the 0.5 threshold in the weighting mechanism? Can you provide an analysis of the impact of using more than one optimization step for the restoration map?

---

### Official Review · Reviewer_6YHo · 2025-10-21

**Soundness:** 2
**Presentation:** 2
**Contribution:** 2
**Rating:** 2
**Confidence:** 3

**Summary:**

The paper presents a simple yet effective test-time defense method with strong empirical validation across diverse medical imaging scenarios. The proposed dynamic weighting mechanism effectively balances adversarial robustness and clean accuracy. However, the mathematical formulation of the restoration process lacks complete derivation and justification. The method demonstrates excellent extensibility to other VLMs and adversarial fine-tuned models.

**Strengths:**

1. TAME exhibits a simple and intuitive design, making it easy to implement and follow.
2. The effective integration of TAME with adversarially fine-tuned models highlights its complementary advantages.
3. Extensive evaluations across 11 medical datasets covering 9 imaging modalities provide compelling evidence of its strong generalizability.

**Weaknesses:**

1. The considered attack methods (PGD, C&W, and AutoAttack) in this paper are white-box, and they are no longer state-of-the-art.
2. The defense assumes that the test image can be modified via known geometric transformations (rotation, cropping, etc.). However, in realistic medical scenarios, the imaging pipeline is fixed, and whether an image has undergone such transformations is often unknown.
3. There is limited discussion on how the choice of transformation strategies influences the overall defense effectiveness.
4. The attack budget $\epsilon_p$ used in the experiments is relatively small (1/255, 2/255, 4/255). In practice, this value can reach up to 16/255, and it remains unclear how TAME performs under larger perturbation budgets.
5. The overall organization of the paper should be improved. Some content, such as Figure 2 and its related descriptions, could be moved to the appendix, while important experimental results, such as the ablation study, should be included in the main text.
6. Minor points:

    - Line 185:  A proper citation should be provided when the abbreviations TTC and R-TPT first appear.

    -  Line 277: The phrase “13 common transformation strategies” should be corrected to “12 common transformation strategies.”

    -  Line 321, Please clarify why the claim that "  $P(\tilde{I})$ approximates $P(\hat{I})$" holds.

**Questions:**

1.  The effectiveness of TAME relies on the divergent responses of clean and adversarial images under small transformations. However, it is unclear whether TAME remains effective when the order of geometric transformation and adversarial attack is reversed (i.e., when the attack is performed after the transformation). In such cases, the adversarial perturbation is optimized over the transformed image space, which may diminish the distributional differences that TAME exploits. The authors are encouraged to discuss or evaluate the robustness of TAME under this setting to better validate its general applicability.
2.   How would TAME perform under black-box  adversarial attacks, which are more realistic for VLMs?
3.  What is the computational overhead compared to standard inference?
4.  How sensitive is the performance to the choice of transformations and the weighting threshold (0.5)?

---

### Official Review · Reviewer_Jhzf · 2025-10-23

**Soundness:** 3
**Presentation:** 2
**Contribution:** 2
**Rating:** 2
**Confidence:** 3

**Summary:**

TAME proposes an innovative testing-time defense framework that effectively enhances the robustness of visual language models in medical image classification tasks by leveraging the semantic vulnerability of adversarial perturbations and the stability differences of the model under slight transformations. This method does not require re-training the model, has low computational costs, and demonstrates significant performance improvements across multiple datasets and attack types, showing good practical potential. However, its core assumption may not be robust in some tasks and modalities, and the theoretical support for the dynamic weight mechanism also needs to be improved. Overall, TAME provides a novel and inspiring idea for adversarial defense research.

**Strengths:**

The TAME method innovatively utilizes the "semantic vulnerability" of adversarial perturbations and proposes the idea of restoring the model's robustness through consistency constraints during the testing phase. This approach has strong theoretical significance.
The method is simple in structure and can significantly improve the model's classification performance under various attacks without the need for re-training, demonstrating good generalization and practical value.

**Weaknesses:**

The core assumption of TAME is not always valid under certain model structures or data modalities, which may lead to misjudgments and "over-correction". Although the dynamic weight mechanism can alleviate this problem to some extent, its calculation method is heuristic and dependent on empirical thresholds, lacking an ablation study for defining the threshold. In addition, the experimental indicators of the paper are relatively simple, mainly based on classification accuracy. Finally, TAME has only been verified in the medical image classification task and lacks generalization verification in more complex tasks such as detection and segmentation. The writing quality is not satisfactory, especially in the contribution section, where it is difficult to understand what the actual contributions of the paper are.

**Questions:**

1. The core assumption of the TAME method is that clean samples should maintain stable prediction results under slight transformations, while attacked samples exhibit significant instability. This assumption has been verified to some extent in experiments. However, its effectiveness highly depends on the model structure and data modality. In such cases, a slight transformation alone can lead to prediction differences, causing the model to mistakenly classify clean samples as "attacked", thereby triggering an erroneous repair process. The authors introduced a dynamic weight mechanism ω. Although ω can partially buffer the negative impacts of the assumption's failure in practice, can it fundamentally resolve the potential conflict between the consistency assumption and data semantics?
2. TAME achieves restoration through a single gradient update. It can be understood that the author chose "one-step restoration" for efficiency, but this might also be a bit crude.
3. The evaluation indicators are limited, and the types of attacks are restricted.
4. The definition of ω seems to be a heuristic ratio. The truncation threshold is set manually and has not been adequately verified for optimality.
5. The authors only consider CLIP-based models and refer to them as visual-language models (VLMs). While CLIP can indeed be regarded as a VLM, the term “VLM” is more commonly used to describe image-to-text models. Therefore, the authors should clearly define the scope of VLMs referred to in their work.
6. The observations may not be generalizable. Additional experiments on other models would help confirm whether the same pattern holds across different architectures.

---

### Official Review · Reviewer_Xs2R · 2025-11-12

**Soundness:** 3
**Presentation:** 2
**Contribution:** 2
**Rating:** 4
**Confidence:** 3

**Summary:**

The paper introduces TAME (The Attack Means Nothing), a simple test-time defense method that enhances the zero-shot adversarial robustness of medical vision-language models like BiomedCLIP without requiring retraining. It builds on the finding that adversarial perturbations are semantically fragile, meaning small transformations can easily disrupt them while leaving clean images mostly unaffected. TAME uses a learnable restoration map to correct adversarial inputs and a dynamic weighting mechanism to avoid harming clean performance. Tested across 11 medical datasets and 9 imaging modalities, TAME consistently outperforms existing defenses, improving robustness by over 47% under various white-box attacks, while maintaining clean accuracy, demonstrating a plug-and-play, efficient, and generalizable approach for robust medical VLM deployment.

**Strengths:**

The paper introduces a novel and practical approach to improving adversarial robustness in medical vision-language models through the idea of semantic fragility—an original insight that reframes test-time defense as a restoration task. The proposed TAME method is technically sound, computationally efficient, and extensively validated across 11 medical datasets and multiple attack settings, showing strong and consistent gains over prior work.

**Weaknesses:**

-- The paper motivates its study around medical imaging but does not clearly justify why the proposed method is specifically relevant to this domain. Since TAME’s formulation does not rely on medical image characteristics, it is unclear whether its advantages stem from domain-specific properties or general VLM robustness. Evaluating on natural image datasets (e.g., ImageNet, CIFAR, or COCO) would help demonstrate that the approach generalizes beyond medical data and clarify its broader impact.

-- All experiments are conducted using a single model, BiomedCLIP, which limits the generality of the findings. It remains unclear whether TAME would show similar robustness improvements on other vision-language models such as CLIP, MedCLIP, or BLIP. Evaluating across multiple architectures would strengthen confidence that the proposed defense is not overly dependent on the specific properties or training data of BiomedCLIP.

-- The inference-time optimization step, although lightweight, may still add latency that could hinder clinical deployment

**Questions:**

-- Can the authors clarify why medical imaging was chosen as the exclusive domain for evaluation, given that TAME’s design does not seem to exploit any domain-specific characteristics? Would similar results hold on natural image datasets such as ImageNet or COCO?

-- Since all experiments are conducted on BiomedCLIP, how well would TAME generalize to other vision-language models (e.g., CLIP, MedCLIP, BLIP)? Are there any dependencies on the architecture or pretraining data that might limit transferability?

-- Have the authors tested TAME under black-box or adaptive attack settings? If not, how might such adversaries affect its performance or reveal limitations of the defense strategy?

---

### Note · Authors · 2025-11-13

I have read and agree with the venue's withdrawal policy on behalf of myself and my co-authors.